# 3D tomographic limb sounder retrieval techniques: irregular grids and Laplacian regularisation

Lukas Krasauskas[1], Jörn Ungermann[1], Stefan Ensmann[1], Isabell Krisch[1], Erik Kretschmer[2], Peter Preusse[1], and Martin Riese[1]

[1]Institute of Energy and Climate Research – Stratosphere (IEK-7), Research centre Jülich GmbH, Jülich, Germany
[2]The Institute of Meteorology and Climate Research (IMK), Karlsruhe Institute of Technology, Karlsruhe, Germany

*Correspondence to:* L. Krasauskas (l.krasauskas@fz-juelich.de)

**Abstract.** Multiple limb sounder measurements of the same atmospheric region taken from different directions can be combined in a 3D tomographic retrieval. Mathematically, this is a computationally expensive inverse modelling problem. It typically requires an introduction of some general knowledge of the atmosphere (regularisation) due to its underdetermined nature.

This paper introduces a consistent, physically motivated (no ad-hoc parameters) variant of the Tikhonov regularisation
scheme based on spatial derivatives of first order and Laplacian. As shown by a case study with synthetic data, this scheme, combined with irregular grid retrieval methods employing Delaunay triangulation, improves both upon the quality and the computational cost of 3D tomography. It also eliminates grid dependence and the need to tune parameters for each use case. The few physical parameters required can be derived from in situ measurements and model data. Tests show that 82% reduction in the number of grid points and 50% reduction in total computation time, compared to previous methods, could be
achieved without compromising results. An efficient Monte Carlo technique was also adopted for accuracy estimation of the new retrievals.

## 1 Introduction

Dynamics and mixing processes in the upper troposphere and lower stratosphere (UTLS) are of great interest. They control the exchange between these layers (Gettelman et al., 2011) and have a strong influence on the composition and thereby on
radiative forcing (Forster and Shine, 1997; Riese et al., 2012). High spatial resolution observations are required to understand the low length-scale processes typical of this region.

Infrared limb-sounding is an important tool for measuring the temperature and volume mixing ratios of trace gases in UTLS with high resolution, especially in the vertical direction, which is critical for resolving typical structures there (e.g. Birner, 2006; Hegglin et al., 2009). Modern limb sounders are capable of high frequency operation resulting in many measurements
taken close to one another, which are best exploited if the data from these measurements can be combined. Producing a 2D curtain of an atmosphere along the flight path of the instrument by combining the observations close to each other is already an established technique (e.g. Steck et al., 2005; Worden et al., 2004). With suitable measurement geometries, this technique can be extended to obtain 3D data.

A tomographic data retrieval uses multiple measurements of the same air mass, taken from different directions, to obtain high resolution 3D temperature and trace gas concentration data. There is, as it often happens with remote sensing retrievals, a large number of atmospheric states that would agree with the given observations within their expected precision. A regularisation algorithm is employed to pick the solution that is in best agreement with our prior knowledge of the atmospheric state and general understanding of the physics involved. The mathematical framework of the tomographic retrieval is outlined in section 2.1.

The classic Tikhonov regularisation scheme (Tikhonov and Arsenin, 1977) is an established technique used for infrared limb sounding instruments such as the Michelson Interferometer for Passive Atmospheric Sounding (Steck et al., 2005; Carlotti et al., 2001)), and GLORIA (Ungermann et al., 2010; Ungermann et al., 2011). It quantifies the spatial continuity of an atmospheric state by evaluating spatial derivatives of the retrieved quantities. This technique serves the purpose of ruling out very pathological, oscillatory retrieval results, but requires fine tuning of several unphysical ad-hoc parameters and subsequent validation. In this paper, we introduce an improved, more physically and statistically motivated approach to regularisation, that requires less tuning. It relies on the calculation of both the first spatial derivative and Laplacian of atmospheric quantities, which provide more complete information about smoothness and feasibility of a particular atmospheric state. The regularisation parameters we use are physical, in the sense that they can be estimated from in situ measurements or model data, and less grid dependent: their values, once established, can be more readily used for new retrievals.

Data retrievals from limb sounders are typically performed on a rectilinear grid. In this paper, we define rectilinear grid by taking a set of longitudes, a set of latitudes and a set of altitudes and placing a grid point at each of the possible combinations of these coordinates. Such grids can be a limiting factor for efficiency of numerical calculations. Due to the exponential nature of atmosphere density distribution with altitude, most of the radiance along any line of sight of a limb imager comes from the vicinity of the lowest altitude point on a given line of sight (called *tangent point*). The resolution of the retrieved data depends on the density of tangent points in the area. For airborne observations this density is highly inhomogeneous. The densely measured regions are limited in size, located below flight altitude only and rarely rectangular. The large, poorly resolved areas with little or no tangent points still need to be included in the grid as long as lines of sight of any measurements pass through them. A rectilinear grid retrieval tends to either underresolve the well-measured area, or waste memory and computation time for regions with few measurements. To avoid this problem, a tomographic retrieval on irregular grid with Delaunay triangulation was developed (section 2). It allows for significant computational cost improvements without compromising retrieval quality.

Most retrieval error estimation techniques would be unreasonably computationally expensive in the case of a 3D tomographic retrieval. Monte Carlo methods allow a relatively quick error estimation. In order to apply Monte Carlo for a 3D tomographic retrieval on an irregular grid and using our newly developed regularisation, dedicated algorithms have to be developed. These are presented in section 3.

The new methods described in sections 2 and 3 were tested with synthetic measurement data to compare their results and computational costs. The tests, and their results, are described in detail in section 4.

## 2 Regularisation improvements and irregular grids

### 2.1 The retrieval

The main focus of this paper is the algorithm for retrieving atmospheric quantities, such as temperature or trace gas mixing ratios, from limb sounder measurements in 3D. In this section, we outline the general inverse modelling approach to this problem and identify some aspects we aim to improve upon.

Let an atmospheric state vector $\boldsymbol{x} \in \mathbb{R}^n$ represent the values of some atmospheric quantities on a finite grid and let $\boldsymbol{y} \in \mathbb{R}^m$ be a set of $m$ remote measurements taken within the region in question. The physics of the measurement process itself has to be understood for the instrument to be practical, so one can usually build a theoretical model of the measurement (the forward model) $F : \mathbb{R}^n \to \mathbb{R}^m$. The inverse problem of determining $\boldsymbol{x}$ given $\boldsymbol{y}$ is then typically both underdetermined and ill-posed: many atmospheric states would result in the same measurement, but due to instrument and model errors no states would yield the exact measurement result $\boldsymbol{y}$. One can solve this problem by finding an atmospheric state $\boldsymbol{x}$ that minimises the following quadratic cost function

$$J(\boldsymbol{x}) = (\boldsymbol{F}(\boldsymbol{x}) - \boldsymbol{y})^T \mathbf{S}_\epsilon^{-1} (\boldsymbol{F}(\boldsymbol{x}) - \boldsymbol{y}) + (\boldsymbol{x} - \boldsymbol{x}_{\mathrm{a}})^T \mathbf{S}_{\mathrm{a}}^{-1} (\boldsymbol{x} - \boldsymbol{x}_{\mathrm{a}}) \tag{1}$$

The first term of (1) quantifies the difference between the actual measurement and a simulated one (given the atmospheric state is $\boldsymbol{x}$). The matrix $\mathbf{S}_\epsilon^{-1} \in \mathbb{R}^m \times \mathbb{R}^m$ represents the expected instrument errors. The second term introduces regularisation, i.e. it is larger for $\boldsymbol{x}$ that are unlikely given our general, measurement unrelated, knowledge about the atmosphere. In practice this is achieved using an a priori state $\boldsymbol{x}_{\mathrm{a}}$, usually derived from climatologies. It is meant for introducing the best of our prior knowledge into the retrieval process without imposing any features of the kind to be measured. The precision matrix $\mathbf{S}_{\mathrm{a}}^{-1}$ (the inverse of a covariance matrix for atmospheric states $\mathbf{S}_{\mathrm{a}}$) contains the prior knowledge about the probability distribution of atmospheric states. For a more detailed discussion of this approach refer to Rodgers (2000).

Finding the precision matrix $\mathbf{S}_{\mathrm{a}}^{-1}$ that would correctly represent the physical and statistical properties of the atmosphere is challenging. The Tikhonov regularisation approach (Tikhonov and Arsenin, 1977)

$$\mathbf{S}_{\mathrm{a}}^{-1} = \alpha_0^2 \mathbf{L}_0 \mathbf{L}_0^T + \alpha_v^2 \mathbf{L}_z \mathbf{L}_z^T + \alpha_h^2 \mathbf{L}_x \mathbf{L}_x^T + \alpha_h^2 \mathbf{L}_y \mathbf{L}_y^T \tag{2}$$

is widely used for this purpose. Here the matrix $\mathbf{L}_0$ is diagonal and contains reciprocals of standard deviations of atmospheric quantities, and $\mathbf{L}_x$, $\mathbf{L}_y$, $\mathbf{L}_z$ represent spatial derivatives in the respective directions on regular rectangular grid, estimated by forward differences. The positive constants $\alpha_0$, $\alpha_h$, $\alpha_v$ are chosen ad-hoc. This is a convenient way to construct $\mathbf{S}_{\mathrm{a}}^{-1}$, and it serves the purpose of ruling out very pathological, oscillatory retrievals. The constants $\alpha_0, \alpha_h, \alpha_v$ are, however, unphysical, grid dependent, and hence have to be estimated (typically by trial-and-error) for each use case. They are not related to the properties of the atmosphere in any simple fashion. Also, the usage of first order derivatives only is justified mostly by simplicity and the need to keep the computations cheap.

We chose a slightly different, more physically and statistically motivated approach to regularisation, which is introduced in the following section.

## 2.2 Covariance

This section describes the theoretical background for the regularisation algorithm used for our retrieval, i.e. the motivation for $\mathbf{S}_{\mathrm{a}}^{-1}$ in equation (1).

If the atmospheric state $\boldsymbol{x}$ has the mean $\boldsymbol{x}_{\mathrm{apr}}$ and covariance matrix $\mathbf{S}_{\mathrm{a}}$ then its entropy is maximised if $\boldsymbol{x}$ has multivariate normal distribution (proved e.g. by Rodgers, 2000). Hence by treating $\boldsymbol{x}$ as a random vector with this distribution we impose the smallest restriction possible while still introducing a priori information from $\mathbf{S}_{\mathrm{a}}$ into our retrieval. Also, the probability density of $\boldsymbol{x}$ is then

$$f_{\boldsymbol{x}}\left(\hat{\boldsymbol{x}}\right) \sim \exp\left(-\left(\hat{\boldsymbol{x}} - \boldsymbol{x}_{\mathrm{apr}}\right)^{T}\mathbf{S}_{\mathrm{a}}^{-1}\left(\hat{\boldsymbol{x}} - \boldsymbol{x}_{\mathrm{apr}}\right)\right) \tag{3}$$

This immediately justifies the second term of equation (1), provided that $\mathbf{S}_{\mathrm{a}}^{-1}$ represents, to some extent, the actual statistics of the atmosphere. This is often referred to as the Bayesian approach to regularisation (Rodgers, 2000). The precision matrix from equation (2), is, however, a mathematical device not meant to represent the physical world, as discussed in the previous section. Furthermore, we would like to derive a coordinate independent expression for the regularisation term of cost function (also unlike equation (2)), as this would be useful for work with irregular grids and allow us to use the same regularisation on completely different grid geometries. To achieve these goals, we use continuous covariance operators and their associated norms and only discretise them as a last step, hence preserving the grid independence and their statistical interpretation. The resulting discrete regularisation is a variant of Tikhonov in its final numerical form, but can be interpreted as a realisation of general continuous covariance relations. The following paragraph introduces some of the required formalism (Lim and Teo, 2009; Tarantola, 2013). Let us denote the departure of the atmospheric quantity $f\left(\boldsymbol{r}\right)$ from the a priori by $\phi\left(\boldsymbol{r}\right) = f\left(\boldsymbol{r}\right) - f_{\mathrm{apr}}\left(\boldsymbol{r}\right)$ for some $r \in \mathbb{R}^{3}$. In some finite volume $V \subset \mathbb{R}^{3}$, we define the covariance operator C by

$$\mathrm{C}\phi\left(\boldsymbol{r}\right) = \int_{\boldsymbol{r}' \in V} \phi\left(\boldsymbol{r}'\right)C_{k}\left(\boldsymbol{r}, \boldsymbol{r}'\right)\mathrm{d}V \tag{4}$$

where $C_{k} : \mathbb{R}^{3} \times \mathbb{R}^{3} \to \mathbb{R}$ is the covariance kernel (also known as covariance function). Then we can treat the scalar fields $\phi\left(\boldsymbol{r}\right)$ as elements of Hilbert space with the product

$$\langle \phi, \varphi \rangle = \int_{V} \phi\left(\boldsymbol{x}\right)\mathrm{C}^{-1}\varphi\left(\boldsymbol{x}\right)\mathrm{d}V \tag{5}$$

which induces the norm $\|\phi\|^{2} = \langle \phi, \phi \rangle$. Once the explicit expression for the norm is found it can be discretised, representing the scalar field $\phi\left(\boldsymbol{r}\right)$ by the atmospheric state vector $\boldsymbol{x} - \boldsymbol{x}_{\mathrm{apr}}$. Then we can write

$$\|\phi\|^{2} = \left(\boldsymbol{x} - \boldsymbol{x}_{\mathrm{apr}}\right)^{T}\mathbf{S}_{\mathrm{a}}^{-1}\left(\boldsymbol{x} - \boldsymbol{x}_{\mathrm{apr}}\right) \tag{6}$$

It now remains to find an appropriate covariance kernel $C_{k}$. Let us first consider an atmosphere that is isotropic and has the same physical and statistical properties everywhere. Then one would expect $C_{k}\left(\boldsymbol{r}, \boldsymbol{r}'\right) = g\left(\|\boldsymbol{r} - \boldsymbol{r}'\|\right)$ with some monotonously decreasing function $g : [0, \infty) \to [0, \infty)$. The most common kernel used in literature for fluid dynamics, meteorology and

similar applications is the parametric Matérn covariance kernel (introduced, e.g. by Lim and Teo (2009))

$$C_\nu\left(\boldsymbol{r},\boldsymbol{r}'\right)=\sigma^2\frac{2^{1-\nu}}{\Gamma\left(\nu\right)}\left(\sqrt{2\nu}\frac{d}{L}\right)^\nu K_\nu\left(\sqrt{2\nu}\frac{d}{L}\right) \tag{7}$$

where $d=\|\boldsymbol{r}-\boldsymbol{r}'\|$, $\sigma$ and $L$ are the standard deviation and typical length scale of structures (correlation length), respectively, of the atmospheric quantity in question. $K_\nu$ is the modified Bessel function of the second kind, $\Gamma(\nu)$ is the Gamma function. Exponential and Gaussian covariance kernels are special cases of the kernel (7), with $\nu=0.5$ and $\nu=1$ respectively. We choose the exponential covariance

$$C_k\left(\boldsymbol{r},\boldsymbol{r}'\right)=\sigma^2\exp\left(-\frac{\|\boldsymbol{r}-\boldsymbol{r}'\|}{L}\right) \tag{8}$$

as it closely resembles the Matérn with $\nu$ values that are typically used for fluid problems and allows for analytic derivation of the subsequently required quantities. Also, having a significant number of parameters to estimate theoretically, we could not make a good use of the flexibility provided by the free parameter of the Matérn covariance in any case.

It can be shown (Tarantola, 2013) that the norm associated with the covariance (8) is

$$\|\phi\|^2=\frac{1}{8\pi\sigma^2}\int\limits_{\boldsymbol{r}\in V}\left[\frac{\phi^2}{L^3}+\frac{2\|\nabla\phi\|^2}{L}+L\left(\Delta\phi\right)^2\right]\mathrm{d}V \tag{9}$$

neglecting some boundary terms. In a more realistic picture of the atmosphere, the correlation length $L$ depends on altitude and strongly depends on direction: correlation between vertically separated air parcels is much weaker than between air parcels separated by the same distance horizontally. We propose to deal with anisotropic or variable $L$ by performing a coordinate transformation such that $L$ would be isotropic and constant in resulting coordinates. In particular, let $U,V\subset\mathbb{R}^3$ and consider a bijective map $\boldsymbol{\xi}:V\to U$ such that both $\boldsymbol{\xi}$ and $\boldsymbol{\xi}^{-1}$ are twice differentiable on their respective domains and have non-zero first and second derivatives everywhere. Then, using integration by substitution and basic vector calculus identities, equation (9) can be written as

$$\|\phi\|^2=\int\limits_{\boldsymbol{u}\in\boldsymbol{\xi}^{-1}(V)}\frac{|\det\left(\mathrm{D}\boldsymbol{\xi}\right)|}{8\pi\sigma^2}\left(\frac{\left(\phi\left(\boldsymbol{\xi}\right)\right)^2}{L^3}+\frac{2\|\boldsymbol{\delta}\left(\phi\left(\boldsymbol{\xi}\right)\right)\|^2}{L}+L\,\mathrm{Tr}^2\left[\left(\mathrm{D}\boldsymbol{\xi}\right)^{-T}\left[\mathrm{D}^2\phi\left(\boldsymbol{\xi}\right)-\boldsymbol{\delta}\left(\phi\left(\boldsymbol{\xi}\right)\right)\cdot\mathrm{D}^2\boldsymbol{\xi}\right]\left(\mathrm{D}\boldsymbol{\xi}\right)^{-1}\right]\right)\mathrm{d}U \tag{10}$$

$$\boldsymbol{\delta}\left(\phi\left(\boldsymbol{\xi}\right)\right)=\left(\mathrm{D}\boldsymbol{\xi}\right)^{-T}\nabla\left(\phi\left(\boldsymbol{\xi}\right)\right) \tag{11}$$

Here $(\mathrm{D}\boldsymbol{\xi})_{ij}=\partial\xi_i/\partial u_j$ is the Jacobian, and we define the matrix $(\mathrm{D}^2 f)_{ij}=(\partial^2 f)/(\partial u_i\partial u_j)$. $\boldsymbol{\xi}$ can be chosen so that all its spatial derivatives could be computed analytically, and $\nabla(\phi(\boldsymbol{\xi}))$, $\mathrm{D}^2(\phi(\boldsymbol{\xi}))$ are the derivatives of $\phi$ in the transformed space, so numerical evaluation of (10) is similar in complexity to that of (9), the only notable difference being the need to compute the mixed derivatives $\partial^2\phi/(\partial u_i\partial u_j)$, $i\neq j$ if $\mathrm{D}\boldsymbol{\xi}$ is not diagonal. A large and relevant class of suitable maps $\boldsymbol{\xi}$ can be defined, for example, by assuming that the correlation lengths in vertical and horizontal direction are smooth functions of altitude $L_v(z)$, $L_h(z)$ (we use Cartesian coordinates $(x,y,z)$ where $z$ axis is vertical). Then the map

$$\boldsymbol{\xi}^{-1}\left(x,y,z\right)=\frac{1}{L}\left(xL_h(z),\,yL_h(z),\,\int\limits_0^z L_v(z')\mathrm{d}z'\right) \tag{12}$$

defines a corresponding transformation. If, in addition, $L_h$ is constant, mixed derivatives are not required and we can simply evaluate $\nabla(\phi(\boldsymbol{\xi}))$, $\Delta(\phi(\boldsymbol{\xi}))$ in transformed space. Transformations of this type could be used, for example, to perform regularisation in tropopause-based coordinates and use different correlation lengths for troposphere and stratosphere. For the study described in this paper, we only introduce anisotropy (i.e. $L_v$ and $L_h$ are different, but do not depend on altitude) and equation (10) becomes

$$\|\phi\|^2 = \frac{1}{8\pi\sigma^2}\left[\int_V \frac{\phi^2 \mathrm{d}V}{L_h^2 L_v} + \frac{2}{L_h}\int_V \frac{L_h}{L_v}\left(\frac{\partial\phi}{\partial x}\right)^2 + \frac{L_h}{L_v}\left(\frac{\partial\phi}{\partial y}\right)^2 + \frac{L_v}{L_h}\left(\frac{\partial\phi}{\partial z}\right)^2 \mathrm{d}V + L_v\int_V \left(\frac{L_h}{L_v}\frac{\partial^2\phi}{\partial x^2} + \frac{L_h}{L_v}\frac{\partial^2\phi}{\partial y^2} + \frac{L_v}{L_h}\frac{\partial^2\phi}{\partial z^2}\right)^2 \mathrm{d}V\right]$$
(13)

This general expression can now be discretised for use in the retrieval. We can represent the scalar field $\phi(\boldsymbol{r})$ by the atmospheric state vector $\boldsymbol{x} - \boldsymbol{x}_{\mathrm{apr}}$ and then calculate a discrete approximation to the integral in (13). This can then be interpreted as $(\boldsymbol{x} - \boldsymbol{x}_{\mathrm{apr}})^T \mathbf{S}_{\mathrm{a}}^{-1}(\boldsymbol{x} - \boldsymbol{x}_{\mathrm{apr}})$ as seen from the equation (6). For example, if a matrix $\mathbf{L}_z$ represents a finite difference scheme to calculate the derivative in z direction, i.e. $(\mathbf{L}_z\boldsymbol{x})_j \approx \frac{\partial\phi}{\partial z}|_{\boldsymbol{r}=\boldsymbol{r}_j}$, and the diagonal matrix $\mathbf{V}$ represents the volumes of grid cells ($\mathbf{V}_{ii}$ being 1/4 of the volume of all grid cells containing grid point $\boldsymbol{r}_i$, see section 2.5 for details), then

$$\sum_i (\mathbf{V}\mathbf{L}_z\boldsymbol{x})_i \approx \int_V \frac{\partial\phi}{\partial z}\mathrm{d}V \qquad \boldsymbol{x}^T\mathbf{L}_z^T\mathbf{V}\mathbf{L}_z\boldsymbol{x} \approx \int_V \left(\frac{\partial\phi}{\partial z}\right)^2 \mathrm{d}V$$
(14)

Other terms of the integral in (13) can be computed similarly. Using the notation of equation (14), with $(\mathbf{L}_{zz}\boldsymbol{x})_j \approx \frac{\partial^2\phi}{\partial z^2}|_{\boldsymbol{r}=\boldsymbol{r}_j}$ and $\mathbf{L}_x$, $\mathbf{L}_y$, $\mathbf{L}_{xx}$, $\mathbf{L}_{yy}$ defined similarly, we construct the precision matrix

$$\mathbf{S}_{\mathrm{a}}^{-1} = \frac{1}{8\pi\sigma^2}\left[\frac{\mathbf{V}}{L_h^2 L_v} + \frac{2}{L_h}\left(\frac{L_h}{L_v}\mathbf{L}_x^T\mathbf{V}\mathbf{L}_x + \frac{L_h}{L_v}\mathbf{L}_y^T\mathbf{V}\mathbf{L}_y + \frac{L_v}{L_h}\mathbf{L}_z^T\mathbf{V}\mathbf{L}_z\right) + L_v\mathbf{L}_\Delta^T\mathbf{V}\mathbf{L}_\Delta\right]$$
(15)

with

$$\mathbf{L}_\Delta = \frac{L_h}{L_v}\mathbf{L}_{xx} + \frac{L_h}{L_v}\mathbf{L}_{yy} + \frac{L_v}{L_h}\mathbf{L}_{zz}$$
(16)

To actually implement the calculation of (15) on an irregular grid one needs to have a triangulation for that grid, an interpolation algorithm (both described in section 2.3). A method to obtain the spatial derivatives of the retrieved quantity at each grid point, needed to obtain $\mathbf{L}_i$, $\mathbf{L}_{ii}$, $i = x, y, z$, is described in section 2.4. The estimation of the physical parameters in equation (13) is discussed in section 4.3. The errors introduced while constructing the precision matrix $S_{\mathrm{a}}^{-1}$ are estimated in Appendix A. Applicability of methods presented in this section to 1D and 2D retrievals is briefly introduced in Appendix B.

## 2.3 Delaunay triangulation and interpolation

We aim to perform a retrieval on an arbitrary, finite grid. This section explains how vertically stretched Delaunay triangulation with linear interpolation is employed for that purpose.

To maintain generality and compatibility with arbitrary grids, we partition the retrieval volume into Euclidean simplexes (tetrahedrons in our 3D case) with vertices at grid points. Many such partitions exist, but it is beneficial to ensure that the

number of very elongated tetrahedrons, that increase interpolation and volume integration errors, are kept to a minimum. The standard technique to achieve this is to use a Delaunay triangulation (Delaunay (1934), Boissonnat and Yvinec (1998)), which maximizes the minimum solid angle of any tetrahedron employed. Recall, also, from section 2.2 that atmospheric quantities in UTLS and the stratosphere above tend to have much more variation vertically than they do horizontally within similar length scales. This difference can be quantified by comparing horizontal and vertical correlation lengths $L_h$ and $L_v$. Hence, stretching the space vertically by means of coordinate transformation $(x, y, z) \mapsto (x, y, \eta z)$ with $\eta \sim L_h/L_v$ before constructing a Delaunay triangulation ensures that, on average, the amount of variation of the retrieved quantity in vertical and horizontal directions is similar. This, in turn, allows us to make use of the aforementioned benefits of Delaunay partitioning. The Computational Geometry Algorithms Library (CGAL, https://www.cgal.org) was used to construct Delaunay triangulations for our grids.

In our implementation, we retrieve several quantities on the same grid, and $L_h$ and $L_v$ may differ for each of them. The precision matrix of equation (15) is evaluated separately for each quantity, so regularisation can still be calculated correctly as long as spatial structures of every quantity are adequately sampled by the grid. In practice, this means that as long as $L_h/L_v$ are of the same order of magnitude for all quantities, they can all be retrieved on one grid. Fortunately this is usually the case (see section 4.3). For test retrievals in this paper $L_h/L_v = 200$ for trace gas concentrations and $L_h/L_v = 67$ for temperature, therefore we set $\eta = 100$. Retrieval results do not indicate systematic oversampling or undersampling in either horizontal or vertical directions and are not sensitive to a change in $\eta$.

In order to keep the computational costs low, we use a simple linear interpolation scheme: in each Delaunay cell we express an atmospheric quantity $f$ at any point $\boldsymbol{x}$ as

$$f(\boldsymbol{r}) = f(\boldsymbol{r_0}) + \boldsymbol{k} \cdot (\boldsymbol{r} - \boldsymbol{r_0}), \quad \boldsymbol{k} = \text{const.} \tag{17}$$

If $\boldsymbol{r}_i$, $0 \le i \le 4$ are the four vertices of the Delaunay cell, the (unique) constant gradient $\boldsymbol{k}$ can obtained from a system of 3 linear equations

$$f(\boldsymbol{r}_i) = f(\boldsymbol{r_0}) + \boldsymbol{k} \cdot (\boldsymbol{r}_i - \boldsymbol{r}_0), \quad 1 \le i \le 3 \tag{18}$$

This method ensures that the interpolated quantity is continuous and consistent with the volume integration scheme described in section 2.5. Implementation is simple and fast, as any point inside a given Delaunay cell can be interpolated with data about that cell only. The gradient $\nabla f$ of the atmospheric quantity, is, however, generally discontinuous at cell boundaries making the interpolation unsuitable for direct use in spatial derivative evaluation.

## 2.4 Derivatives

In order to evaluate the cost function based on 3D exponential covariance (13, 15), we need to estimate $\nabla\phi$ and $\Delta\phi$ at every grid point (as before, $\phi = f - f_{\text{apr}}$ is the departure of atmospheric quantity $f$ from the a priori), i.e. construct the matrices $\mathbf{L}_i, \mathbf{L}_{ii}, i = x, y, z$ of equation (15). This section describes the algorithm we use to achieve that on irregular Delaunay grids.

We begin by establishing some requirements that our derivative-estimation algorithm will have to meet. Let us write $\mathbf{S}_a^{-1} = \mathbf{A} + \mathbf{B}$, where $\mathbf{A}$ is a diagonal matrix representing the first integral of (13) (or first term of (10)), and $\mathbf{B}$ represents the

remaining terms. If $\mathbf{B}$ were an exact representation, it would be positive definite by construction, but this may not hold with $\nabla\phi$ and $\Delta\phi$ obtained numerically for a finite grid. If indeed an atmospheric state vector $\boldsymbol{q}$ exists so that $\boldsymbol{q}^T\mathbf{B}\boldsymbol{q}=0$, then for every atmospheric state $\boldsymbol{x}$

$$(\boldsymbol{x}\pm\boldsymbol{q}-\boldsymbol{x}_{\mathrm{apr}})^T\mathbf{B}(\boldsymbol{x}\pm\boldsymbol{q}-\boldsymbol{x}_{\mathrm{apr}})=(\boldsymbol{x}-\boldsymbol{x}_{\mathrm{apr}})^T\mathbf{B}(\boldsymbol{x}-\boldsymbol{x}_{\mathrm{apr}})\pm 2(\boldsymbol{x}-\boldsymbol{x}_{\mathrm{apr}})^T\mathbf{B}\boldsymbol{q} \tag{19}$$

hence one of the states $\boldsymbol{x}\pm\boldsymbol{q}$ would be favoured (have a smaller cost function) over $\boldsymbol{x}$ by the inverse modelling algorithm. Therefore, a scaled version of $\boldsymbol{q}$ would appear on any retrieval as noise. This particular type of noise would only be suppressed by the cost function term from $(\boldsymbol{x}-\boldsymbol{x}_{\mathrm{apr}})^T\mathbf{A}(\boldsymbol{x}-\boldsymbol{x}_{\mathrm{apr}})$. This guarantees only that $\boldsymbol{x}$ is not very far from a priori, but does nothing to ensure that it is even continuous. This often results in retrievals with unphysical periodic structures (noise) and is unacceptable.

We deal with this problem by explicitly ensuring that $\mathbf{B}$ is positive definite. Consider the estimation of derivatives at a single grid point $\boldsymbol{a}\in\mathbb{R}^3$. $\nabla\phi(\boldsymbol{a})$ and $\Delta\phi(\boldsymbol{a})$ are numerically estimated from the values of $\phi$ in some (say $m$) grid points near $\boldsymbol{a}$. We can write this as a map

$$D_{\boldsymbol{a}}(\phi):\mathbb{R}^m\to\mathbb{R}^6, \quad \{\phi(\boldsymbol{r}_i)-\phi(\boldsymbol{a}):1\le i\le m\}\mapsto\left\{\frac{\partial\phi}{\partial x},\frac{\partial\phi}{\partial y},\frac{\partial\phi}{\partial z},\frac{\partial^2\phi}{\partial x^2},\frac{\partial^2\phi}{\partial y^2},\frac{\partial^2\phi}{\partial z^2}\right\} \tag{20}$$

Now $\boldsymbol{q}^T\mathbf{B}\boldsymbol{q}=0$, $\boldsymbol{q}\ne\boldsymbol{0}$ implies that at every grid point $\boldsymbol{a}\in\mathbb{R}^3$ we have $\nabla\phi(\boldsymbol{a})=0$, $\Delta\phi(\boldsymbol{a})=0$ for the $\phi(\boldsymbol{a})$ corresponding
to state $\boldsymbol{q}$. Hence such $\boldsymbol{q}$ will not exist if we require that $D_{\mathbf{a}}$ have trivial null space for all grid points $\boldsymbol{a}$ (i.e. $D_{\mathbf{a}}(\phi)=\{0,\dots,0\}$ only if $\phi(\boldsymbol{r}_i)=\phi(\boldsymbol{a})$, $1\le i\le m$). The converse is not always true, so the trivial null space requirement is a stronger condition than absolutely necessary. It is, however, advantageous because it restricts the derivative-estimation algorithm at each grid point independently and hence can be implemented without computationally expensive operations on $\mathbf{S}_a^{-1}$ as a whole.

Since (20) will be used directly to construct $\mathbf{S}_a^{-1}$, which does not depend on atmospheric state $\boldsymbol{x}$ and hence on $\phi(\boldsymbol{a})$, $D_{\boldsymbol{a}}$
must be a linear map, so null space is a vector space of dimension $\max\{0,m-6\}$, hence we need $m\le 6$. We also want to avoid the derivative estimation to be underdetermined ($m\ge 6$), so we choose $m=6$, i.e. we aim to construct a linear map (20) that estimates derivatives at grid point $\boldsymbol{a}$ using atmospheric quantity values at 6 points around $\boldsymbol{a}$. This prevents us from using interpolation for derivative calculations, since each interpolated value depends on atmospheric quantity values at multiple (in our case 4) grid points and consequently, even if the same grid point is reused for several interpolated values, the total number
of grid points employed exceeds 6 in any interpolation-based scheme we could come up with.

A solution satisfying the above criterion and able to deal with grid points with neighbours at irregular positions is based on polynomial fitting. For 6 grid points around a grid point $\boldsymbol{a}=(x_a,y_a,z_a)$ denoted as $\boldsymbol{r}_i=(x_a+x_i,x_a+y_i,x_a+z_i)$, $1\le i\le 6$ we write

$$\phi(\boldsymbol{r}_i)-\phi(\boldsymbol{a})=\phi_x x_i+\phi_y y_i+\phi_z z_i+\frac{\phi_{xx}}{2}x_i^2+\frac{\phi_{yy}}{2}y_i^2+\frac{\phi_{zz}}{2}z_i^2, \quad 1\le i\le 6 \tag{21}$$

We solve this as a linear system for the unknowns $\phi_x,\phi_y,\phi_z,\phi_{xx},\phi_{yy},\phi_{zz}$ which can then be directly identified with the derivatives we seek. Note that, as we required, the system can be solved by inverting a 6-by-6 matrix that only depends on $\boldsymbol{r}_i$ and not $\phi(\boldsymbol{r}_i)$, so that we could construct the precision matrix (15) to be used for every atmospheric state on given grid.

The only remaining issue is selecting a suitable set of grid points $r_i$ so that the resulting matrix that needs to be inverted would not be singular or have a very high condition number. We found that this is much easier to achieve by imposing simple geometric criteria for point selection rather than algebraic conditions.

Let $a$ be a point on Delaunay grid. The main idea of the algorithm we propose is to pick, for each spatial dimension $d$, a pair of points $r_1$, $r_2$ suitable for estimating the derivative in that direction. Points of such a pair should be sufficiently separated in $d$ direction (i.e. projections of the vectors $r_i - a$ to $d$ direction sufficiently different), be on the opposite sides of $a$ (unless $a$ is at the boundary of the grid and this is not possible), and be reasonably close to $a$. We obtain the required 6 points by choosing one pair of points for each of the 3 dimensions. This can be implemented as follows.

Let $a = (x_a, y_a, z_a)$, and let us write the coordinates of other points on this grid as $r_i = (x_a + x_i, y_a + y_i, z_a + z_i)$ with $r_i = \|r_i - a\|$. Let $0 < \beta < 1$ and $\gamma > 1$ be constants, suitable numeric values will be discussed further on. Then:

1. For each Delaunay grid neighbour $r_i$ of $a$, compute $\alpha_{ix} = x_i/r_i$, $\alpha_{iy} = y_i/r_i$, $\alpha_{iz} = z_i/r_i$.

2. If $i, j$ such that $|\alpha_{ix}| > \beta$, $|\alpha_{jx}| > \beta$, $\alpha_{ix}\alpha_{jx} < 0$ exist, then pick $r_i$, $r_j$ for derivative calculation. Else if $k, l$ such that $|\alpha_{kx}| > |\alpha_{lx}| > \beta$, $\alpha_{kx}/\alpha_{lx} > \gamma$ exist, then pick $r_l$, $r_k$ for derivative calculation.

3. Repeat step 2 for y and z dimensions. Each point can be only picked for one of the dimensions.

4. If neither condition of step 2 is satisfied for at least 1 dimension, repeat steps 1-3 with not only direct Delaunay grid neighbours of $a$, but also second neighbours (i.e. neighbours of neighbours). If this fails as well, do not calculate derivatives at the point $a$ and simply assume them to be zero.

We found that the polynomial interpolation method is robust and a rather low value of $\beta = 0.3$ was sufficient to produce suitable sets of neighbouring points (i.e. almost all 6-by-6 matrices resulting from such choice of points can be reliably inverted for polynomial fitting). In practice this means that the algorithm can find a suitable pair of points for dimension $d$ unless all the Delaunay neighbours of the point $a$ are located very close to the other two axes w.r.t. the point $a$.

## 2.5   Volume integration

Computing (13) requires a way to estimate volume integrals on Delaunay grid, which is described in this section.

Let a Delaunay cell have vertices $r_i$ and face areas $S_i$ for $0 \le i \le 3$, and volume $V$. Then integrating an atmospheric quantity $f$ as interpolated in (17) over $V$ we get

$$\int_V f(r)\,dV = \frac{V}{4}\sum_{i=0}^{3} f(r_i) \tag{22}$$

One can prove this by writing (17) with $r_i$, $0 \le i \le 3$ instead of $r_0$ and adding up the resulting 4 equations to obtain

$$f(r) = \frac{1}{4}\sum_{i=0}^{3} f(r_i) + k \cdot \left(r - \frac{1}{4}\sum_{i=0}^{3} r_i\right) \tag{23}$$

Now choose Cartesian coordinates $(x', y', z')$ where $\boldsymbol{r_0} = (H, 0, 0)$ and $\boldsymbol{r}_i$, $1 \le i \le 3$ lie in the plane $x' = 0$ and consider the $x'$ component of the integral $\boldsymbol{I} = \int_V \left( \boldsymbol{r} - 1/4 \sum_{i=0}^3 \boldsymbol{r}_i \right) \mathrm{d}V$

$$I_{x'} = \int\limits_V \left( x' - \frac{H}{4} \right) \mathrm{d}V = \int\limits_0^H x' S_0 \left( \frac{H - x'}{H} \right)^2 \mathrm{d}x' - \frac{HV}{4} = \frac{H^2 S_0}{12} - \frac{HV}{4} = 0 \tag{24}$$

As we can choose such coordinates for any vertex instead of $\boldsymbol{r}_0$, the integral $\boldsymbol{I}$ has zero component in 3 linearly independent
directions, so $\boldsymbol{I} = 0$. Then integrating (23) over $V$ yields (22) and completes the proof.

We can now see that (22) gives an exact value to the volume integral of $f$ provided that $f$ behaves exactly as prescribed by our interpolation scheme. Since the interpolation is linear we may expect the values of volume integrals to be correct to the first order in cell dimensions. Since, generally, all data fed into the inverse modelling problem is interpolated beforehand, there is little reason to expect that a more elaborate and accurate volume integration scheme would improve the accuracy of retrievals
on its own. A method based on Voronoi cells was also attempted and did not produce meaningfully different results.

## 3  Monte Carlo diagnostics

Estimating the precision of remote sensing data products generated by means of inverse modelling is essential for the users of the final data and also valuable for evaluation and optimisation of the inverse modelling techniques. Detailed quantitative descriptions of data accuracy can be derived in theory (see validation in section 4.6 and Rodgers (2000)) but they are, in case of
large retrievals, too numerically expensive to calculate in practice. The accuracy calculation typically involves matrix inversion and hence is slower than $O\left(n^{2.8}\right)$ operations (Skiena, 1998) for an atmospheric state vector of $n$ elements (for 3D tomography $n$ might be of the order $10^6$). One way of reducing computational cost is is employing the Monte Carlo technique to generate a set of random (Gaussian) atmospheric state vectors with the required covariance. One can then add these random vectors to the result of the retrieval and run the forward model on the perturbed atmospheric states. The magnitude of perturbations that
is required to produce a variation of forward model output similar to the expected error of measurements would then be an estimate of the retrieval error (Ungermann et al., 2011).

In order to implement this technique, one needs a way to generate a random vector $\boldsymbol{x}$ with the required covariance, represented by a known, symmetric, positive definite covariance matrix $\mathbf{S}_{\mathrm{a}}$. This can be done by generating a vector $\boldsymbol{u}$ of independent standard normal random variables and finding a square matrix $\tilde{\mathbf{L}}$ such that $\tilde{\mathbf{L}}\tilde{\mathbf{L}}^T = \mathbf{S}_{\mathrm{a}}$ (Kroese et al., 2013). Then we can set
$\boldsymbol{x} = \tilde{\mathbf{L}}\boldsymbol{u}$, and indeed $\langle \boldsymbol{xx}^T \rangle = \langle \tilde{\mathbf{L}}\boldsymbol{uu}^T \tilde{\mathbf{L}}^T \rangle = \tilde{\mathbf{L}} \langle \boldsymbol{uu}^T \rangle \tilde{\mathbf{L}}^T = \mathbf{S}_{\mathrm{a}}$, i.e. the vector $\boldsymbol{x}$ has the required covariance (the angle brackets in the expression above denote the expected value).

A widely used technique for obtaining the matrix $\mathbf{L}$ is the Cholesky decomposition (Golub, 1996). Given the covariance matrix $\mathbf{S}_{\mathrm{a}} = \{s_{ij}\}$ it explicitly provides a lower triangular matrix root $\tilde{\mathbf{L}} = \{l_{ij}\}$ satisfying $\tilde{\mathbf{L}}\tilde{\mathbf{L}}^T = \mathbf{S}_{\mathrm{a}}$ as shown in (25).

$$l_{ij} = \begin{cases} \sqrt{s_{ij} - \sum_{k=1}^{i-1} l_{ik}^2}, & i = j \\ l_{jj}^{-1} \left( s_{ij} - \sum_{k=1}^{j-1} l_{ik} l_{jk} \right), & i > j \\ 0, & i < j \end{cases} \tag{25}$$

In practice we do not usually assemble the covariance matrix $\mathbf{S}_a$, but rather its inverse: the precision matrix $\mathbf{S}_a^{-1}$, because the latter is sparse. This is not an issue, since $\mathbf{S}_a^{-1}$ is then also symmetric positive definite, so one can compute its root $\mathbf{L}$ such that $\mathbf{L}\mathbf{L}^T = \mathbf{S}_a^{-1}$ in the same way as above, and then obtain $\boldsymbol{x}$ from the linear system $\mathbf{S}_a^{-1}\boldsymbol{x} = \mathbf{L}\boldsymbol{u}$. We will refer to the components of precision matrix by $\mathbf{S}_a^{-1} = \{a_{ij}\}$.

Cholesky decomposition does not preserve the sparse structure of $\mathbf{S}_a^{-1}$, i.e. the $\mathbf{L}$ obtained in this way will typically have many more non-zero entries than $\mathbf{S}_a^{-1}$. It follows from (25), however, that if $\mathbf{S}_a^{-1}$ has lower half-bandwidth $w$ (definition in equation (26) below), then so has $\mathbf{L}$

$$w = \min\{k \geq 0 : i - j > k \Rightarrow a_{ij} = 0\} \tag{26}$$

In practice, this means that if $\mathbf{S}_a^{-1}$ is a sparse $N \times N$ matrix with half-bandwidth $w \ll N$, $\mathbf{L}$ will have approximately $N(w+1)$
non zero entries. Cholesky decomposition is hence well suited for computing diagnostics of 1D retrieval: assuming that the value of the retrieved atmospheric parameter at one point only directly correlates with its $w$ nearest neighbouring points in each direction, one gets a precision matrix with half-bandwidth $w$ and can compute $\mathbf{L}$ cheaply with $O(Nw^2)$ operations.

The situation is very different in the higher dimensions. We will use some results of graph theory to show that Cholesky decomposition is not practical in those cases. The $n$-dimensional lattice graph $P_k^n$ consists of an $n$-dimensional rectangular
grid of size $k$ in each direction, with edge between any two grid neighbours. Let us label the vertices of $P_k^n$ with integers: $P_k^n = \{v_i : 1 \leq i \leq k^n\}$, and let $q$ be the largest difference of indexes of adjacent vertices ($q = \max\{|i-j| : v_i, v_j \text{adjacent vertices}\}$). Then the *bandwidth* $\varphi(P_k^n)$ of $P_k^n$ is defined as the minimum possible value of $q$ among all possible ways to label the vertices. Now say we have some physical quantity defined on each vertex of this grid. Then any reasonable precision matrix $\mathbf{S}_a^{-1}$ for this grid would at least give non-zero correlations between grid neighbours, i.e. $\left(\mathbf{S}_a^{-1}\right)_{ij} \neq 0$ if $v_i$ and $v_j$ are neighbours. Then,
by comparing the definitions of $\varphi(P_k^n)$ and lower-half bandwidth $w$ of the precision matrix from the last paragraph, one can see that $2w + 1 \geq \varphi(P_k^n)$. FitzGerald (1974) showed that $\varphi\left(P_k^2\right) = k$ and $\varphi\left(P_k^3\right) = \lfloor 3k^2/4 + k/2 \rfloor$. Therefore, the narrowest possible half-bandwidths of $\mathbf{S}_a^{-1}$ are $w = O(k)$ in 2D and $w = O\left(k^2\right)$ in 3D. It follows that if the grid contains a total of $N$ points, the computational cost of Cholesky decomposition would be $O(N^2)$ in 2D and $O\left(N^{7/3}\right)$ in 3D, which is unsatisfactory for large retrievals.

For these higher dimensions, we need to use sparse matrix iterative techniques to reduce computational cost and memory storage requirements, such as Krylov subspace methods. In general, it is rather difficult to find a simple iteration scheme that would compute a square root of a matrix and converge reasonably fast. Here we will follow an algorithm proposed by Allen et al. (2000). Consider a system of linear ordinary differential equations (ODEs) with initial condition

$$\begin{cases} \mathrm{d}\boldsymbol{v}/\mathrm{d}t = -\frac{1}{2}\left(\mathbf{S}_a^{-1}t - (1-t)I\right)^{-1}\left(\mathbf{I} - \mathbf{S}_a^{-1}\right)\boldsymbol{v}(t) \\ \boldsymbol{v}(0) = \boldsymbol{u} \end{cases} \tag{27}$$

where $\boldsymbol{v}(t)$ is a column vector of size $N$, $\mathbf{I}$ is the identity matrix and $\mathbf{S}_a^{-1}$ is a $N \times N$ symmetric positive definite (s. p. d.) matrix, prescaled so that $\|\mathbf{S}_a^{-1}\|_\infty < 1$ (i. e. $\mathbf{S}_a^{-1} - \mathbf{I}$ is non-singular). $\left(\mathbf{S}_a^{-1} - \mathbf{I}\right)$ and $\left(\mathbf{S}_a^{-1}t + (1-t)\mathbf{I}\right)^{-1}$ commute for

$0 \leq t \leq 1$, which makes it easy to verify that

$$\boldsymbol{v}(t) = \left(\mathbf{S}_{\mathrm{a}}^{-1}t - (1-t)\,\mathbf{I}\right)^{1/2}\boldsymbol{u} \tag{28}$$

is the solution of (27). Hence we can obtain $\boldsymbol{v}(1) = \mathbf{S}_{\mathrm{a}}^{-1/2}\boldsymbol{u}$ by solving (27) numerically, and then solve the linear system $\mathbf{S}_{\mathrm{a}}^{-1}\boldsymbol{x} = \boldsymbol{v}(1)$ for the vector $\boldsymbol{x}$, so that $\boldsymbol{x} = \mathbf{S}_{\mathrm{a}}\mathbf{S}_{\mathrm{a}}^{-1/2}\boldsymbol{u}$. The matrix $\mathbf{S}_{\mathrm{a}}^{-1/2}$ above is symmetric by construction (sum of products of symmetric matrices), hence

$$\langle \boldsymbol{x}\boldsymbol{x}^T \rangle = \mathbf{S}_{\mathrm{a}}\mathbf{S}_{\mathrm{a}}^{-1/2}\langle \boldsymbol{u}\boldsymbol{u}^T \rangle \mathbf{S}_{\mathrm{a}}^{-1/2}\mathbf{S}_{\mathrm{a}} = \mathbf{S}_{\mathrm{a}} \tag{29}$$

i.e. $\boldsymbol{x}$ is indeed the vector required for Monte Carlo simulations. Note also that one does not need to explicitly calculate matrix inverses while solving the ODE, since

$$\left(\mathbf{S}_{\mathrm{a}}^{-1}t - (1-t)\,\mathbf{I}\right)\frac{\mathrm{d}\boldsymbol{v}}{\mathrm{d}t} = -\frac{1}{2}\left(\mathbf{I} - \mathbf{S}_{\mathrm{a}}^{-1}\right)\boldsymbol{v}(t) \tag{30}$$

is a linear system that can be solved for $\mathrm{d}\boldsymbol{v}/\mathrm{d}t$.

We chose a classical approach – a Runge-Kutta method, to solve the linear ODE system numerically. Using constant step size proved to be inefficient, so adaptive step size control was introduced (in particular, a fifth order Runge-Kutta-Fehlberg method by Fehlberg (1985)). Conjugate gradients method (Saad, 2003) was employed for all the linear equation systems involved. If we make the same assumption about the structure of the inverse of the correlation matrix as before, i.e. that atmospheric quantities are only correlated between nearest grid neighbours, the said matrix will have a constant number of non-zero entries in each row for any amount of rows (grid points) $N$. More explicitly, under this assumption the covariance matrix for 1D retrieval would be tridiagonal, and for a 3D retrieval on a regular rectangular grid it would contain 6 non-zero entries in each row. Therefore, each iteration of the conjugate gradients algorithm will cost $O(N)$ operations. It is difficult, in general, to estimate the dependence of required number of iterations on $N$, but in our case large matrices seem to converge similarly to smaller ones and the whole computational cost of generating random vectors remains close to $O(N)$, which is a huge improvement over the Cholesky decomposition. The algorithm does not, though, explicitly yield the root of the covariance matrix and only gives one random vector per run. Hence, unlike the Cholesky decomposition, it must be executed again for each new random vector required, so it is only feasible if the number of random vectors required is a lot smaller than $N$. Fortunately, this is very much the case in practice, since the number of random vectors required is typically only of the order of 100 (Ungermann, 2013).

Although this root-finding algorithm does not provide a way to directly multiply the matrix root with itself, there is an inexpensive way to verify that the matrix root was computed correctly. Let $\mathbf{S}$ be a $n \times n$ s. p. d. matrix, let $\boldsymbol{e}_i$ be the $i$'th unit vector $(1 \leq i \leq n)$, and compute $\boldsymbol{v}_i = \mathbf{S}^{1/2}\boldsymbol{e}_i$ using our algorithm. Then for any $i,j$ we have $\boldsymbol{v}_i^T\boldsymbol{v}_j = \boldsymbol{e}_i^T\left(\mathbf{S}^{1/2}\right)^T\mathbf{S}^{1/2}\boldsymbol{e}_i = \boldsymbol{e}_i^T\mathbf{S}\boldsymbol{e}_i = \mathbf{S}_{ij}$, an element of $\mathbf{S}$. Hence the quality of the matrix root can be evaluated by comparing $\boldsymbol{v}_i^T\boldsymbol{v}_j$ with $\mathbf{S}_{ij}$. The error tolerances of conjugate gradients and Runge-Kutta were adjusted so that these values would differ by at most $10^{-4}$ when factoring a matrix prescaled so that the largest eigenvalue is approximately 0.95.

## 4 Performance study

### 4.1 The GLORIA instrument and data processing

The Gimballed Limb Observer of Air Radiance in the Atmosphere (GLORIA) is an aircraft-based infrared limb imager. It is a Michelson interferometer with a 2D detector array, spectral coverage from 770 to 1400 $cm^{-1}$ and spectral sampling of up to 0.0625 $cm^{-1}$ (Riese et al. (2014); Friedl-Vallon et al. (2014)). It is carried aboard the Russian M55 Geophysica or German HALO research aircraft, and is generally intended to look to the side of the aircraft, but can pivot horizontally so that the angle between aircraft heading and observation direction can be changed from $45°$ to $135°$. The limb observation setup naturally allows high vertical resolution (up to approximately 200 m) and, with a proper trade-off between spectral resolution and measurement time, a high density of vertical profiles sampled along-track. The pivoting allows to look at the same air mass from several points along the flight path and use tomography to improve resolution in the horizontal direction perpendicular to the flight path. Best results with tomographic retrievals are achieved, however, when the aircraft flies in a path close to circular (hexagonal flight paths are used in practice, so that the aircraft could fly straight most of the time) of around 400 km in diameter, and GLORIA observes the air masses in the middle from many directions (Ungermann et al., 2011). A horizontal resolution down to 25 km in both directions can be achieved this way. As any limb observer, GLORIA can only provide tangent point data about air masses at and below aircraft flight altitude, which is limited to, approximately 20 km for the M55 Geophysica and 15 km for HALO. For more detail on GLORIA, refer to e.g. Friedl-Vallon et al. (2014).

The implementations of the algorithms described in the sections 2 and 3 were integrated into the Jülich Rapid Spectral Simulation Code Version 2 (JURASSIC2). The forward model of atmospheric radiance used in this code employs a forward model for atmospheric radiation based on radiances obtained from the emissivity growth approximation method (Weinreb and Neuendorffer (1973), Gordley and Russell (1981)) and the Curtis–Godson approximation (Curtis (1952), Godson (1953)). For more information about JURASSIC2, refer to Hoffmann et al. (2008); Ungermann et al. (2011); Ungermann (2013). The performance of the new algorithms, both in terms of output quality and computational cost, is evaluated in the rest of this section.

### 4.2 Retrieval setup and test data

A hexagonal flight pattern intended for tomography was realised on 25th of January 2016, as part of flight 10 of the POL-STRACC measurement campaign. The flight path of the HALO research aircraft contained a regular hexagon with diameter (distance between the opposite vertices) of around 500 km over Iceland, which allowed for a high spacial resolution retrieval in the central part of the aforementioned hexagon (Figure 1). The flight altitude throughout the hexagonal flight segment remained close to 14 km. For detailed information about this flight refer to Krisch et al. (2017).

The test case used in this paper is based on the actual aircraft path, measurement locations, spectral lines used for retrieval and meteorological situation during this flight. Synthetic measurement data was used instead of real GLORIA observations: the forward model of JURASSIC2 was employed to simulate the observed radiances in the atmospheric state given by the ECWMF temperature and WACCM model data for trace gases, simulated instrument noise was subsequently added to those radiances.

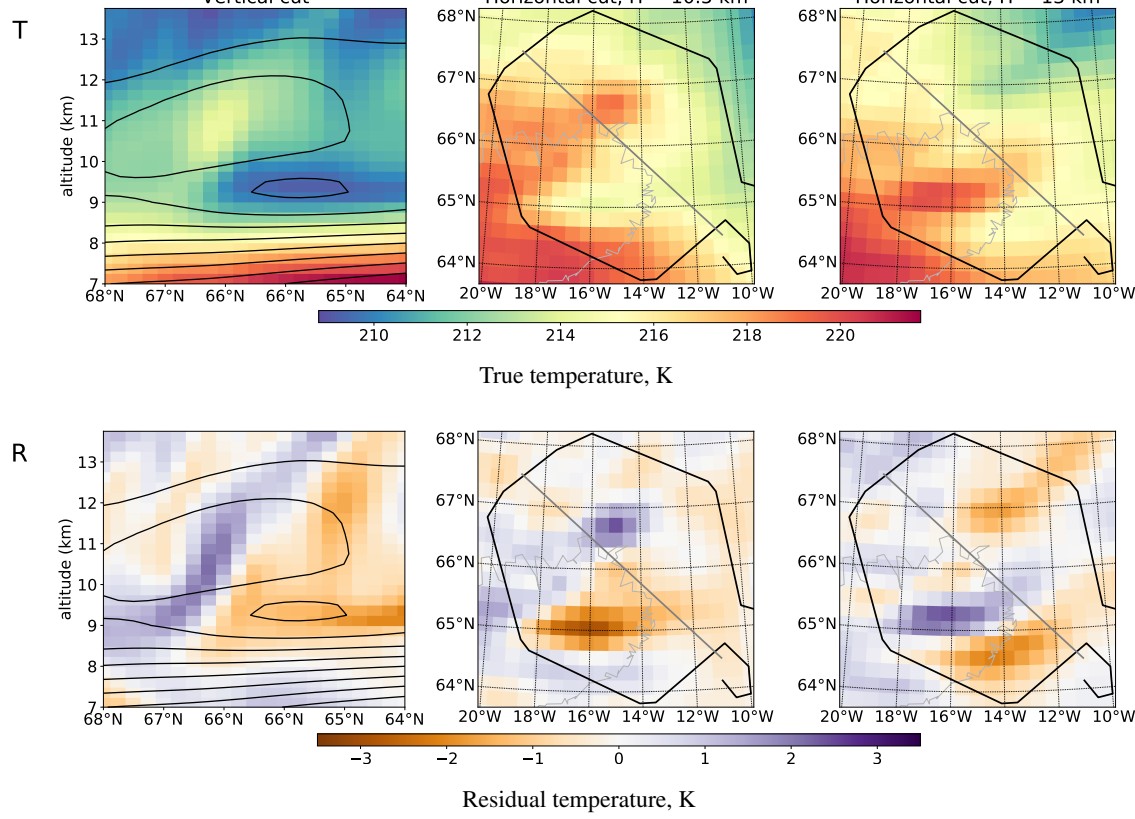

**Figure 1.** The first row (T) shows the ECMWF temperature data used to generate the synthetic measurements (true temperature). Second row (R) shows the temperature residual: the difference between the true temperature and the smoothed temperature field used as an a priori. In the first column, black lines represent contours of a priori temperature value. In columns 2 and 3: black line – flight path, thick grey line – position of cut shown in the first column, thin grey line – topography (Icelandic coastline visible).

This setup allows us to use model data as a reference (the "true" atmospheric state) for evaluation of retrieval quality. The same set of simulated measurements was used for all test retrievals described in this paper. These measurements were obtained by running the forward model on a very dense grid (about twice as dense in each dimension as those of the densest test retrievals). This was done to ensure that the discretisation errors in the simulated measurements would be minimal and would not favour any retrieval (as it may could happen, if they were generated on the same grid). An evaluation of forward model errors on different grids is presented in Appendix C.

The retrieval derives temperature and volume mixing ratios of $O_3$, $HNO_3$, $CCl_4$, and $ClONO_2$. Temperature and $CCl_4$ concentration are derived in the altitude range from 3 km to 20 km; $O_3$, $HNO_3$, $ClONO_2$ are retrieved from 3 km to 64 km. The latter three trace gases have larger volume mixing ratios above flight level and hence significant contributions to measured radiances from these high altitudes. Interpolated WACCM data was used as a priori for all trace gases. A priori data

**Table 1.** Retrieval parameters

| Entity | $\alpha_0$ | $\alpha_h$ | $\sigma$ |
|---|---|---|---|
| Temperature | $10^{-3}$ | $10^{-1}$ | 0.645 K |
| $O_3$ | $10^{-5}$ | $10^{-3}$ | 70.7 ppbv |
| $HNO_3$ | $10^{-5}$ | $10^{-3}$ | 902 pptv |
| $CCl_4$ | $10^{-5}$ | $10^{-3}$ | 2.13 pptv |
| $ClONO_2$ | $10^{-5}$ | $10^{-3}$ | 76.7 pptv |

Regularisation weights $\alpha_0$, $\alpha_h$, (as in (2)) are used for retrieval A, the standard deviation $\sigma$ (see (15)) used for retrievals B-D

for temperature was obtained by applying polynomial smoothing to ECMWF data. The retrieval is hence not supplied with the full temperature structure that was used to simulate the measurement. Therefore, the agreement between the retrieval result and the "true" atmospheric state, upon which the simulated measurements are based, cannot be achieved by overregularisation.

### 4.3 Correlation length estimation

The standard deviation $\sigma$ and correlation the lengths $L_h$ and $L_v$ in equation (13) can be obtained from statistical analysis of in situ measurement or model data. Rigorous derivation of these parameters for each retrieved trace gas and temperature would require detailed analysis of in situ measurement databases and is, therefore, out of scope of this paper. However, the following rough approximation proved sufficient to improve upon retrieval results compared to the old regularisation scheme.

  3D tomographic retrievals are most useful for those trace gases that have high vertical gradients and complex spatial struc-
ture. The spatial correlation lengths of concentrations of such gases are mostly determined by stirring, mixing and other dynamical processes and are therefore similar. Furthermore, Haynes and Anglade (1997) and Charney (1971) estimated the dynamically determined aspect ratio to be $\alpha \approx 200 - 250$ in lower stratosphere, and we expect less than that in the upper troposphere. Hence we will assume $L_h/L_v = 200$ for all trace gases.

  An analysis of spatial variability of some airborne in situ measurements was performed by Sparling et al. (2006). For ozone
concentration $\chi(s, z)$, where $s$ is the horizontal location and $z$ is the altitude, they define fractional difference parameters

$$\Delta_{r,h} = \frac{2 \left| \chi(s+r, z+h) - \chi(s, z) \right|}{\left[ \chi(s+r, z+h) + \chi(s, z) \right]} \tag{31}$$

$$\Delta_r = \lim_{h \to 0} \Delta_{r,h} \tag{32}$$

and provide the dependence of the standard deviation $\sigma(\Delta_r)$ on the horizontal separation $r$. If the typical spatial vari-
ation in ozone concentration is significantly smaller than the mean ozone concentration $\langle \chi \rangle$, one can approximate $\Delta_r \approx \left| \chi(s+r) - \chi(s) \right| / \langle \chi \rangle$. Then, using the covariance relation (8) and various properties of the normal distribution, we get

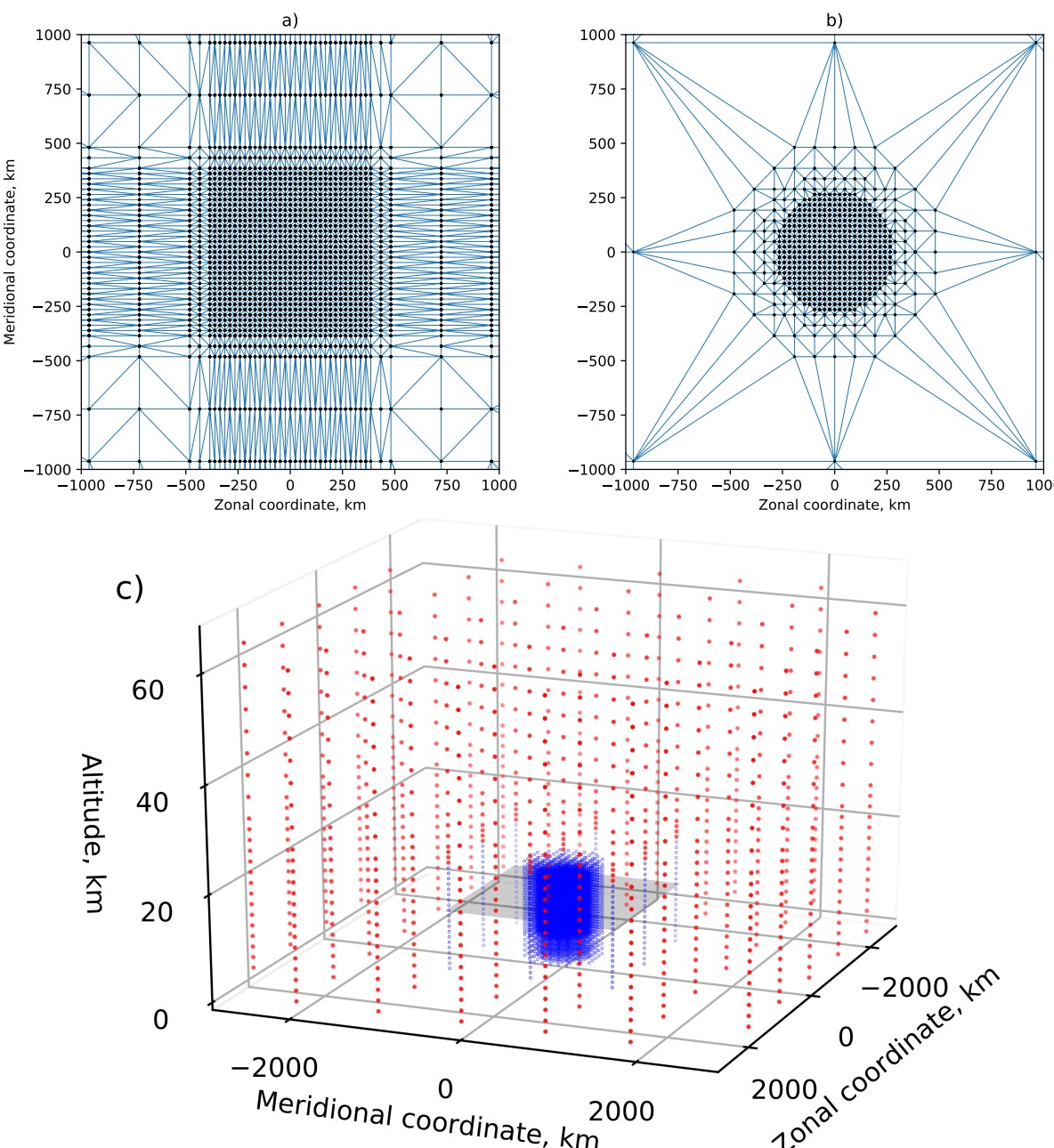

**Figure 2.** Panels (a) and (b): horizontal cuts of retrieval grids. Dots show grid points, blue lines indicate Delaunay cell boundaries. (a) shows the full initial grid for retrievals A, B, C (all horizontal layers of this grid are identical), (b) shows the thinned out grid for retrieval D (cut at 11 km). Panel (c): grid for retrieval D. Radial distance is defined as distance to the vertical line at the centre of the hexagonal flight path (66°N, 15°W). Points within the inner core (1000 km × 1000 km × 20 km) are shown in blue, other points – in red, shaded rectangle shows the position of horizontal cut in panel (b).

$\sigma^2(\Delta_r) = 2\left(\sigma/\langle\chi\rangle\right)^2\left(1 - 2/\pi\right)\left(1 - \exp\left(-r/L_h\right)\right)$. By comparing this to experimental results of Sparling et al. (2006), we estimate $L_h = 200\,\mathrm{km}$. Using the aspect ratio argument above, we also set $L_v = 1\,\mathrm{km}$.

The spatial structure of retrieved temperature differs from that of the trace gases. It is determined not only by mixing and stirring, but also radiative processes and gravity waves. Radiative transfer tends to erase some of the fine vertical structure

determined by isentropic transport, hence one should use a greater vertical correlation length than in the case of trace gases. Gravity waves have a variety length scales, but due to the finite resolution of the instrument not all of them can be retrieved. Following the gravity wave observational filter study for the GLORIA instrument in Krisch et al. (2018) we can estimate lowest observable gravity wave horizontal wavelength to be $\lambda_h \sim 200\,\mathrm{km}$ and lowest vertical wavelength $\lambda_v \sim 3\,\mathrm{km}$. $\lambda_h$ coincides with the dynamically determined correlation length $L_h = 200\,\mathrm{km}$, so this value should be appropriate for temperature as well.

The $\lambda_v$ limit is not only determined by instrument resolution, but is also related the fundamental properties of gravity waves. Preusse et al. (2008) gives the following expression for the gravity wave saturation amplitude

$$\hat{T}_{\max} = \frac{\lambda_v \bar{T} N^2}{2\pi g} \tag{33}$$

where $\bar{T}$ is the mean temperature and $N$ is the Brunt–Väisälä frequency. In typical lower stratosphere conditions and $\lambda_v = 3\,\mathrm{km}$ the equation (33) gives $\hat{T}_{\max} \approx 4\mathrm{K}$. Since gravity waves are usually not saturated and considering typical measurement error

(see 4.6), waves of significantly shorter wavelengths would probably not be observed. We hence set $L_v = 3\,\mathrm{km}$. Time series of measurements at a fixed point in atmosphere are then used to approximate the standard deviations $\sigma$ for all retrieved quantities (Table 1). Our choice of $\sigma$ value for temperature, in particular, may seem rather low. This is related to the choice of a priori: smoothed ECMWF data was used instead of climatologies that would have been more typical in this case. Such an a priori can be expected to match the large scale structures of real temperature more closely, thus reducing the forward model errors

in poorly resolved areas and improving the results. It is particularly useful for resolving fine structures, such as gravity waves, which were the main scientific interest of the measurment flight used as a basis for test retrievals (Krisch et al., 2017). A likely closer match between a priori and retrieval thus requires a lower $\sigma$ value, that, in this setup, represents expected strength of gravity wave disturbances, rather than full thermal variability of the atmospheric region in question.

## 4.4   Test retrievals

The test data described in the previous section was processed with several different regularisation setups and retrieval grids. Here we present the results for four different runs.

Firstly, as a reference, the latest version of JURRASIC2 without the implementations of any new algorithms described in this paper was used (retrieval A). It uses a rectangular grid that covers altitudes from 1 km to 64 km. The vertical spacing of the grid is 1 km between the altitudes of 1 km and 5 km, 250 m between altitudes of 5 km and 20 km, 1 km between altitudes

of 20 km and 25 km and 4 km between 28 km and 64 km. As required for the rectilinear grid, the number and distribution of points is the same in each altitude (Figure 2a). For this reference run, a first order regularisation (1) with trilinear interpolation (Bai and Wang, 2010) was used. The regularisation weights, as defined in (2), are given in Table 1. These are typically tuned ad-hoc, i.e. adjusted by trial-and-error until optimal retrieval results can be achieved, starting from the default values obtained

**Table 2.** Grid densities in different regions for retrieval D in kilometers

| Radius | <300 | 300-400 | 400-550 | 550-1500 | >1500 |
|---|---|---|---|---|---|
| Altitude | Horizontal grid separation | | | | |
| 1-5 | 100 | 100 | 100 | 1000 | 1000 |
| 5-12.5 | 25 | 50 | 100 | 1000 | 1000 |
| 12.5-14.5 | 25 | 25 | 100 | 1000 | 1000 |
| 14.5-17 | 25 | 50 | 100 | 1000 | 1000 |
| 17-20 | 100 | 100 | 100 | 1000 | 1000 |
| 20-64 | 500 | 500 | 500 | 1000 | 1000 |
| Altitude | Vertical grid separation | | | | |
| 1-5 | 1 | 1 | 1 | 1 | 2 |
| 5-12.5 | 0.125 | 0.25 | 1 | 1 | 2 |
| 12.5-14.5 | 0.125 | 0.125 | 1 | 1 | 2 |
| 14.5-18 | 0.125 | 0.25 | 1 | 1 | 2 |
| 18-24 | 2 | 2 | 2 | 2 | 2 |
| 24-64 | 4 | 4 | 4 | 4 | 4 |

as in Hansen and O'Leary (1993). While tuning, retrieval results are validated against the atmospheric state used to generate the synthetic measurements. When using real observations, results are validated against in situ measurements and model data.

Then, to evaluate the performance of second order regularisation, retrieval B was performed. The grid and interpolation methods were identical to retrieval A, but the regularisation was replaced with the second order scheme from equations (13), (15) and correlation lengths derived in section 4.3 from in situ observations. No subsequent tuning of these parameters was performed.

We compare the quality of different retrievals by inspecting the differences between the retrieved temperature and temperature used to generate the synthetic measurements ("true" temperature). These differences are shown on horizontal and vertical slices of the observed atmospheric volume in Figure 3. Figure 4 shows the variation of the retrieved and true temperatures on horizontal lines in order to visualise the response of retrievals to different spatial structures in true temperature.

Comparison of Figure 3 rows A and B shows the effect of the new regularisation (13). The new algorithm (retrieval B) demonstrates better agreement with the true temperature. It shows markedly less retrieval noise in the central area, as well as better results (less effect of a priori) just outside it, at the edges of the vertical cut shown in Figure 3. Considering also that, unlike A, the regularisation strength of B was not manually tuned for this test case in particular, we can conclude that the new regularisation performed better in this test case.

The third retrieval (C) was performed on the same set of grid points as B and with the same input parameters, but the grid was treated as irregular, i.e. a Delaunay triangulation was found, the algorithms described in section 2 were used for regularisation.

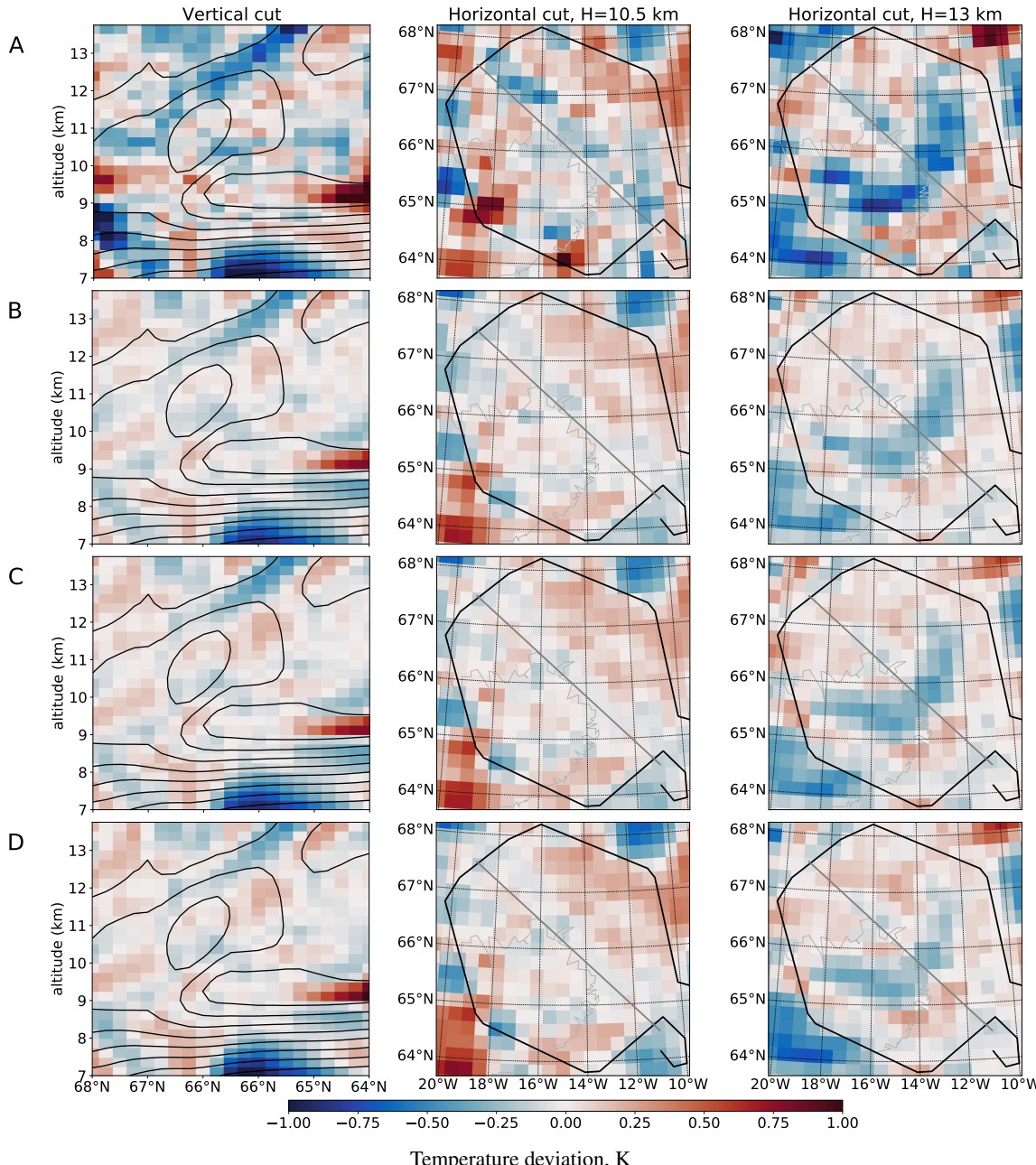

**Figure 3.** Panels show difference between retrieved temperature and true temperature of the simulated atmosphere. Rows A-D show results of respective retrievals. In the first column, black lines represent contours of a priori temperature value. In columns 2 and 3: black line – flight path, thick grey line – position of cut shown in the first column, thin grey line – topography (Icelandic coastline visible).

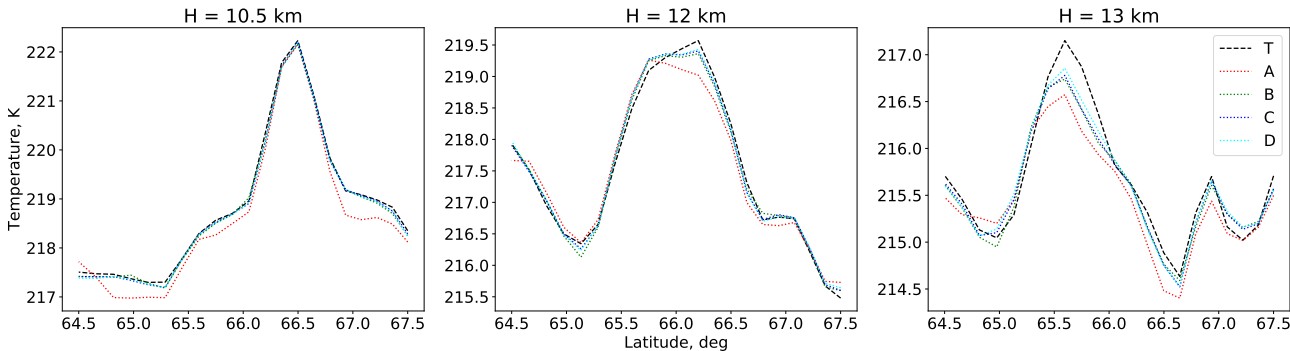

**Figure 4.** Retrieved temperature at different altitudes in the vertical cut shown in Figures 1 and 3. Line T represents the true temperature of the simulated atmosphere, A-D: respective retrievals.

A horizontal cut of the Delaunay triangulation for this grid is shown in Figure 2a. By comparing B and C one can evaluate the agreement between the old interpolation, derivative calculation and volume integration techniques for rectilinear grids and their newly developed irregular-grid-capable alternatives. The temperature fields from retrievals B and C are very similar (Figure 3), which confirms that the new Delaunay triangulation based code gives consistent results when run on rectilinear grid.

Finally, retrieval D was performed with the same input parameters and methods as C, but using only the subset (22 %) of the original grid points to reduce computational cost. These points were chosen so that grid density would be highest in the volumes best resolved by GLORIA and sparser where little measurement data is available. In particular, a set of axially symmetric regions around the center of original retrieval (i.e. symmetric w.r.t. vertical axis at 66°N, 15°W) was defined as shown in Table 2. Points were removed from the original grid to achieve the specified vertical and horizontal resolutions

for each region, resulting in the grid shown in Figure 2. Results changed little compared to retrieval C. Most of the minor differences between C and D occur near the edges of the hexagonal flight path (Figure 3). Even though the grid spacing for retrieval D is still the same as C in some these areas, larger grid cells nearby have some effect on their neighbours. Note also that C and D retrievals tend to differ in the areas where the agreement between each of them and the true temperature is relatively poor and diagnostics predict lower accuracy (Figure 5), so D can be considered to be as good as C in well-resolved

areas. Unphysical temperature structures far outside the measured area like the one partially seen at 64°N, 20°W are not unique to this retrieval, they just occur a little closer to the flight path in this case due to the coarser grid outside the hexagon.

## 4.5 Computational costs

JURASSIC2 finds an atmospheric state minimising (1) iteratively. In each iteration it first computes a Jacobian matrix of the forward model, hence obtains a Jacobian of the cost function (1) and then minimises the cost function using the Levenberg-

Marquardt (Marquardt, 1963) algorithm and employing conjugate gradients (CG) (Saad (2003), Hanke (1995)) to solve the

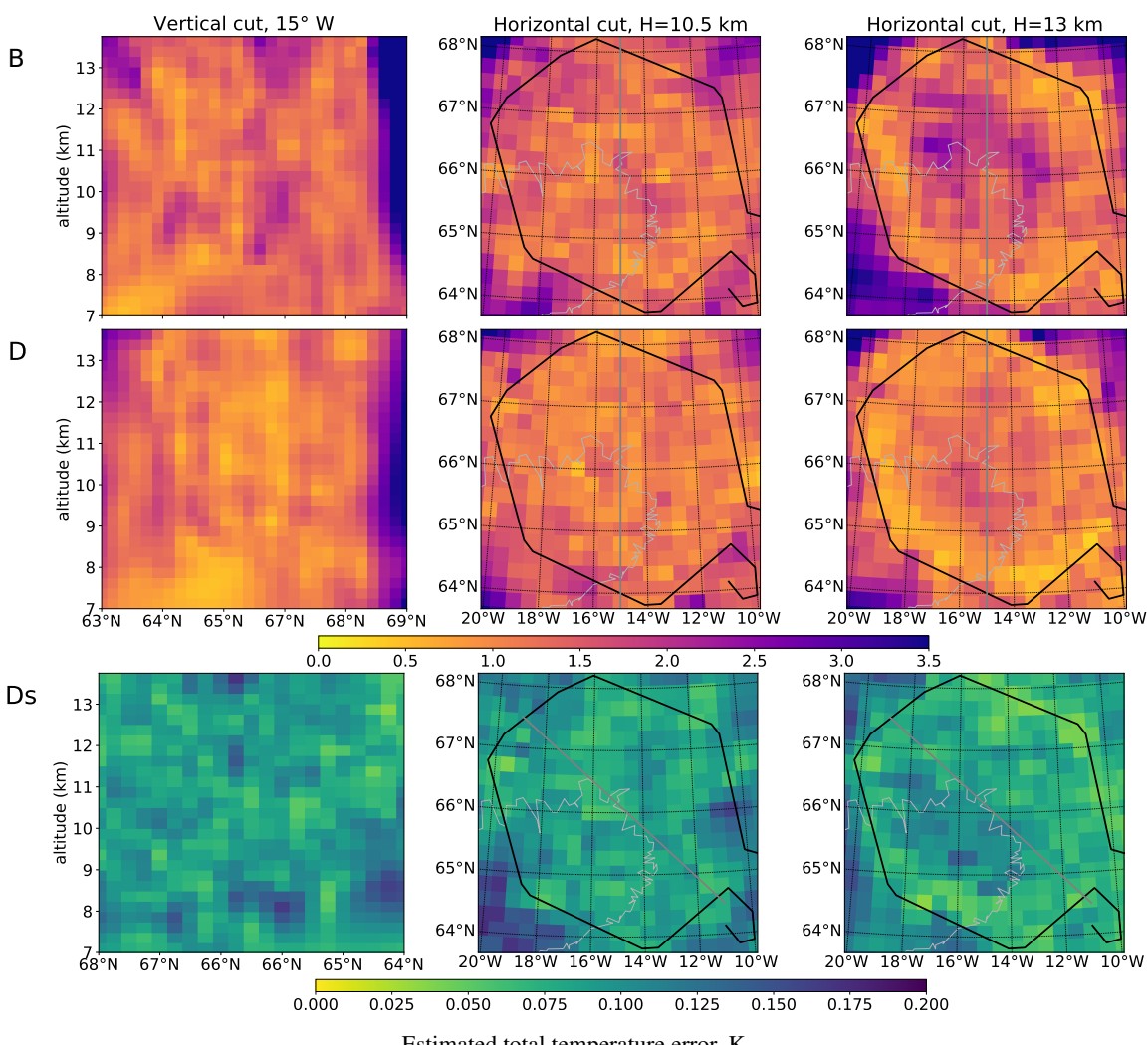

**Figure 5.** Panels B and D show the Monte Carlo estimate of the total temperature error the retrievals B and D would have, if the measurements were real. Panel Ds shows an error estimate for the synthetic retrieval D (this assumes that unretrieved quantities, e. g. pressure, are known exactly). In columns 2 and 3: black line – flight path, thick grey line – position of cut shown in the first column, thin grey line – topography.

linear systems involved. The forward model is then run to simulate the measurements for the new iteration of atmospheric state.

All calculations were performed on the Jülich Research on Exascale Cluster Architectures (JURECA) supercomputer oper-
ated by the Jülich Supercomputing Centre (Jülich Supercomputing Centre, 2016). Each retrieval was executed in parallel on 4
5  computation nodes, each with 24 CPU cores (twin Intel Xeon E5-2680 v3 CPU's) and 128 GB of system main memory.

Table 3 shows the time required for different parts of calculation averaged over 3 identical runs (although < 1% variation was observed). The "Other calculations" field mostly includes time required by input and output, but also preparatory steps, such as Delaunay triangulation in the case of retrievals C and D.

The results in Table 3 reveal the impact of the new methods on computation times. The new, more complex regularisation
results in slower convergence in the conjugate gradient calculation, making retrieval B about 6% more expensive than A. The Delaunay triangulation, however, decreases Jacobian calculation costs by roughly 25%, even the one used on the same grid as the old approach. This has to do with the fact that interpolation based on Delaunay triangulation only requires 4 values of the interpolated quantity at neighbouring points, and linear interpolation in three directions requires 8, thus resulting in more non-zero terms in the Jacobian. The time required for initialisation of Delaunay interpolation and derivatives, including construction
of the matrix $\mathbf{S}_a^{-1}$, increases rather fast with grid size with current implementation, contributing to "other calculations" of retrieval C. Further optimisation of this step is possible and will be implemented in the future. In total, the new retrieval (C) is 9% faster then the old approach (A) while using the same grid.

**Table 3.** Computational cost of retrievals (calculation times given in minutes)

| Retrieval | A | B | C | D |
|---|---|---|---|---|
| Number of grid points | 302526 | 302526 | 302526 | 54580 |
| Jacobian calculation | 134.4 | 133.2 | 101.2 | 61.3 |
| Minimisation (CG) | 4.2 | 14.9 | 12.8 | 6.0 |
| Forward model | 2.5 | 2.5 | 2.6 | 2.6 |
| Other calculations | 5.6 | 4.4 | 17 | 3.0 |
| Total elapsed time | 146.7 | 155.0 | 133.6 | 72.9 |

Retrieval D gives a further 45% reduction of computation time by removing 82% of points in the grid. A speedup of such order was expected. Only a minority of grid points contribute to a forward model (and hence Jacobian) calculations, and
JURASSIC2 code is heavily optimised for sparse matrix operations. Most of the grid points contributing to forward model are in the areas well resolved by the instrument and thus cannot be removed from the grid without compromising data products. Therefore, the actual savings in computation time mostly come from removing points above flight level and choosing a more appropriate non-rectangular shape for the densest part of the grid when measurement geometry requires that. Some of the points above flight path must be included since infrared radiation from there is measured by the instrument, but very little
information about atmosphere far above flight level can be retrieved, so very low grid densities are sufficient (Table 2).

## 4.6 Diagnostics

We follow Ungermann et al. (2011) and use Monte Carlo approach to obtain diagnostics, but replace Cholesky decomposition with the method developed in section 3. Results for retrievals B and D are presented in Figure 5 B, D. They show the total error estimate for temperature based on the sensitivity of temperature value to (simulated) instrument noise and all other

retrieved atmospheric quantities as well as expected error of non-retrieved quantities (e.g. pressure). These results tell us what temperature errors we could expect if the retrievals in question were performed with real data. The panel Ds shows an error estimate based on (simulated) instrument noise and uncertainties of retrieved parameters only. In the case of a synthetic retrieval, unretrieved parameters, upon which the simulated measurements are based, are known exactly. Therefore, panel Ds
is a direct Monte Carlo prediction of the temperature deviation we see in Figure 3D, and the magnitude of this deviation is indeed similar.

These results confirm that temperature can only be reliably retrieved within the flight hexagon or very close to the actual path. Also, the error estimates are higher in the central area of hexagon close to flight altitude, in agreement with test results (compare column 3 of Figures 3 and 5).

Monte Carlo retrieval results provided here were verified by estimating temperature error at one grid point using a more direct approach. Rodgers (2000) proves that retrieved atmospheric state $x$ can be written in terms of the a priori state $x_{\mathrm{a}}$, the (unknown) true state $x_t$ and measurement error $\epsilon$ as

$$x = \mathbf{A}x_{\mathrm{t}} + (\mathbf{I} - \mathbf{A})\,x_{\mathrm{a}} + \mathbf{G}\epsilon \tag{34}$$

where

$$\mathbf{G} = \mathbf{M}^{-1}\left(\mathrm{D}\mathbf{F}(x)\right)^{T}\mathbf{S}_{\epsilon}^{-1} \qquad \mathbf{A} = \mathbf{G}\,\mathrm{D}\mathbf{F}(x) \qquad \mathbf{M} = \mathbf{S}_{\mathrm{a}}^{-1} + \left[\mathrm{D}\mathbf{F}(x)\right]^{T}\mathbf{S}_{\epsilon}^{-1}\mathrm{D}\mathbf{F}(x) \tag{35}$$

Here D denotes the Jacobian operator and all other notation as in (1). The so-called *gain matrix* $\mathbf{G}$ fully describes the effect of measurement uncertainties $\epsilon$ on the retrieved state $x$. Computing $\mathbf{G}$ would be prohibitively expensive due to the required matrix inversion, but note that $i$'th element of $x$ only depends on $i$'th line of the matrix $\mathbf{G}$, so one can use Krylov subspace methods (e.g. conjugate gradients) to solve $\mathbf{M}r_i = e_i$ for the $i$'th row $r_i^{T}$ of the inverse matrix $\mathbf{M}^{-1}$. The $i$'th row of $\mathbf{G}$ can
then be obtained as $r_i^{T}\left(\mathrm{D}\mathbf{F}(x)\right)^{T}\mathbf{S}_{\epsilon}^{-1}$.

This approach allows for error estimation for a small number of grid points in reasonable time. We chose one point in the centre of the hexagon (66° N, 15° W, 11.75 km altitude) for retrieval B. The total temperature error was found to be 1.27 K and agrees with Monte Carlo result (1.25 K) at this point within 2%, which is an excellent accuracy for an error estimate.

## 5   Conclusions

A new regularisation algorithm for 3D tomographic limb sounder retrievals employing second spatial derivatives (based on equations (13), (15)) was developed and implemented. The ad-hoc parameters (regularisation weights) of the previous first spatial derivative based approach were replaced with physical quantities – retrieved quantity standard deviation and correlation lengths in vertical and horizontal directions. Unlike the regularisation weights, these quantities do not require as manual tuning for each retrieval and can be estimated from in situ measurements, model data and theoretical considerations. Tests with
synthetic data showed slightly superior performance of the new algorithm, even compared to the old approach with tuned parameters. A rather general technique (equation (10)) of adapting the tomographic retrieval for atmospheric regions of high anisotropy and spatial variability was also proposed.

A Delaunay triangulation based approach for retrievals on irregular grids was developed. By better adapting the grid to the retrieval geometry and thinning it out in the regions where lack of measurement data limits the resolution, the computation time of a tomographic retrieval was reduced by a total of 50% without significant deterioration of the results. Irregular grids can be further employed for efficient tomographic retrievals in non-standard measurement geometries. At the time of writing, the methods developed in this paper were already in use to process GLORIA limb sounder data.

Monte Carlo Diagnostics were newly implemented for a 3D tomographic retrieval allowing for quick and reliable evaluation of measurement quality, identification of reliably resolved volumes and selection of optimal 3D tomography setups. Error estimates for retrieved atmospheric quantities can now be calculated at each grid point, while the previous approaches to diagnostics only allowed to do that for a few selected points within similar time frame.

## Appendix A: Regularisation tests

The equation (15) was used to construct precision matrices $\mathbf{S}_a^{-1}$ directly, since constructing a covariance matrix $\mathbf{S}_a$ explicitly and inverting it would be prohibitively computationally expensive. While the equation (13) was derived analytically, its discrete implementation cannot be exact, which raises the question whether our precision matrices still have required properties, namely, whether they are still inverses of $\mathbf{S}_a$ that obeys equation (8). Clearly, this can only be verified directly for a small test case, where covariance matrices are small enough to be inverted. Also, employing matrix norms to compare matrices is not very useful in this case, as the directly computed $\mathbf{S}_a^{-1}$ is based on finite difference derivative estimates which do not work very well at grid boundaries. This is not a problem for the retrieval, which, by design, approaches a priori near grid boundaries, but it has strong effects on matrix norms. Instead, we compared the covariance matrices by using them to compute the norms of several test vectors, that represent typical structures expected for that covariance.

Precision matrix $\mathbf{S}_a^{-1}$ was constructed for one entity on a regular 20x20x20 grid (grid constant 1 in each direction) using the same techniques as for retrieval B ($\sigma = 1$ and $L_v = L_h = L = 2$). Then a covariance matrix was explicitly constructed for the same grid using equation (8) and inverted to obtain the precision matrix $\hat{\mathbf{S}}_a^{-1}$. A set of vectors $\boldsymbol{x}_\phi$ – discrete representations of a "Gaussian wave packet" disturbance

$$\phi(\boldsymbol{r}) = \exp\left(-\frac{\|r\|^2}{d^2}\right)\cos(\boldsymbol{k} \cdot \boldsymbol{r}) \tag{A1}$$

was computed. The parameter $d$ was chosen so that the amplitude of the wave would decay to at most 0.01 at the grid boundary, thus suppressing edge effects. Two different wavelengths, namely $\lambda = 15, 20$, in grid units, and were chosen and 50 $\boldsymbol{k}$ values pointing at every possible direction were generated for each wavelength (i.e. $\|\boldsymbol{k}\| = 2\pi/\lambda$). Then, for each of the resulting 100 disturbances the norms $\|\boldsymbol{x}_\phi\|_p$ and $\|\boldsymbol{x}_\phi\|_t$, where $\|\boldsymbol{x}_\phi\|_p^2 = \boldsymbol{x}_\phi^T \mathbf{S}_a^{-1} \boldsymbol{x}_\phi$ and $\|\boldsymbol{x}_\phi\|_t^2 = \boldsymbol{x}_\phi^T \hat{\mathbf{S}}_a^{-1} \boldsymbol{x}_\phi$, were calculated, as well as their relative difference $\delta = 2 \left|\|\boldsymbol{x}_\phi\|_p - \|\boldsymbol{x}_\phi\|_t\right| / \left(\|\boldsymbol{x}_\phi\|_p + \|\boldsymbol{x}_\phi\|_t\right)$. For the case $\lambda = 15$ we found that the mean $\delta$ value was 0.050, with standard deviation 0.0042, and for the case $\lambda = 20$ mean was 0.036, with standard deviation 0.0007. We believe that precision of this order is sufficient for regularisation purposes.

**Table A1.** Forward model tests

| Measurement set | original + noise | A | C | D |
|---|---|---|---|---|
| $\Delta\left(\boldsymbol{y}, \boldsymbol{y}_{\mathrm{orig}}\right)$ | 0.0193 | 0.0035 | 0.0036 | 0.0047 |

Difference in forward model results, compared to original synthetic measurement set
without noise. A-D refer to forward model results of respective retrievals

## Appendix B:  Applicability to 1D and 2D retrievals

The exponential covariance relation, given in equation (8), can be trivially extended to any dimension. It is used as a statistical model to many processes. The norm associated with this exponential covariance, given in equation (9), is, however, a 3D-specific result. Known 2D equivalents involve fractional powers of derivatives and therefore are not as easy to implement numerically. A norm associated with 1D exponential covariance can be shown (Tarantola, 2013) to have the simple form

$$\|\phi\|^2 = \frac{1}{2\sigma^2 L}\left(\int\limits_{x_1}^{x_2}\phi^2 \mathrm{d}x + L^2 \int\limits_{x_1}^{x_2}\left(\frac{\partial\phi}{\partial x}\right)^2 \mathrm{d}x\right) \tag{B1}$$

neglecting some boundary terms. Here the scalar field $\phi(x)$ represents the difference between the quantity to be retrieved and a priori, and $L$ is the correlation length. All the main theoretical results and numerical methods of this paper can therefore be reproduced for 1D inverse modelling problems. The usefulness of our methods for practical 1D applications ultimately depends on the size of the problem in question. 1D limb sounding retrievals, for example, are usually too small for the methods developed here to be useful. It would typically be possible, and indeed simpler, to explicitly construct and invert the covariance matrix of a 1D problem, rather than discretise the associated norm (B1). In the case of a very large 1D problem with strong covariance, however, one may find the methods presented in this paper more lucrative.

## Appendix C:  Forward model tests

As pointed out in section 4.2, the same measurement set, synthetically generated on dense grid, was used for retrievals A-D. This provides an easy way to verify if the forward model is performing consistently on different grids: run it on each grid, based on the "true" atmospheric state and compare the results with the synthetic measurements from the dense grid. In our test case, the radiances are calculated for $n = 31$ spectral windows. If vector $\boldsymbol{y}$ represents a set of radiances, and $\boldsymbol{y}_i$ represents the radiances at $i$-th spectral window, we evaluate the difference between those vectors with the parameter $\Delta\left(\boldsymbol{y}, \boldsymbol{y}'\right) = (1/n)\sum_i^n 2\|\boldsymbol{y}_i - \boldsymbol{y}'_i\|_2/(\|\boldsymbol{y}_i\|_2 + \|\boldsymbol{y}'_i\|_2)$ where $\|\cdot\|_2$ is the euclidean norm. The results of this comparison are presented in Table A1. Forward model for retrievals A and C used the same grid, but A used trilinear interpolation, and C used Delaunay interpolation. We can see that the change in interpolation does not introduce significant errors (compared to those introduced by the grid change). Thinned out grid and Delaunay interpolation was used for grid D. Note that the simulated measurements that were actually used also have simulated instrument noise added to them. The results clearly show that the

effects of different discretisation are well below instrument noise level. Also, the error is lower than our generally assumed forward model error of 1-3% (Hoffmann, 2006).

*Acknowledgements.* The POLSTRACC campaign was supported by the German Research Foundation (Deutsche Forschungsgemeinschaft, DFG Priority Program SPP 1294). The authors gratefully acknowledge the computing time granted through JARA-HPC on the supercomputer JURECA at Forschungszentrum Jülich. The European Centre for Medium-Range Weather Forecasts (ECMWF) is acknowledged for meteorological data support. The authors especially thank the GLORIA team and the POLSTRACC flight planning and coordination team for their great work during the POLSTRACC campaign on which the test cases in this paper are based.

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
