# Peer review of "3D tomographic limb sounder retrieval techniques: irregular grids and Laplacian regularisation"

_Atmospheric Measurement Techniques, 2018_

## Referee Comment (RC1) · Anonymous Referee #1 · 24 Sep 2018

Review of: Tomographic airborne limb sounder retrievals on irregular grid with second order regularisation

by Krasauskas and colleagues.

High level comments:

This is a very nice, well-written paper that describes thorough work to develop and evaluate new approaches to the 3D-retrieval problem from airborne tomographic limb sounding observations. Although the paper is dense and mathematical, this is driven by the subject matter and, in my view, in line with what is required for completeness. I'm very happy to recommend that this paper be accepted with only minor revisions

detailed below.

My only high-level concern is that the end results, based on simulations for GLORIA, do not necessarily show the instrument or technique off at its best. In part, as I see it, this is because of the tight nature of the a priori constraints imposed, specifically the 0.645 K a priori precision estimate for temperature. As an aside, although the choice of the associated correlation lengths is well described in the manuscript, I was unable to find discussion of how the 0.645K number was arrived at (apologies if I missed it). A consequence of this very tight constraint is that the instrument has to "work very hard" to add useful information, as witness the similarity between the magnitude of the fields in figures 6 and 4 to those in the bottom row of figure 3. In other words, the instrument hasn't been "allowed" to tell you that much more than you already knew.

Some limb sounding teams opt to use very loose a priori constraints to enable the retrieved precision (Sx in the Rodgers notation) to be a clear marker of where information came from the instrument and what from the a priori (and/or use the averaging kernels to the same effect). While I recognize that this is not ideal, and that, indeed, having a more geophysically informed choice of a priori description is one of the cornerstones of your work, it would perhaps enable you to better quantify the strengths of your new technique compared to the prior state of the art. To be honest, 0.645 K feels tight even for a "geophysically informed" value of the a priori precision. It's possible I've understood some aspect of the work in this discussion, so apologies if so.

I'm not necessarily advocating a reappraisal or rerun of the work here, merely suggesting that the authors consider this in future work, and/or possibly include some brief discussion of the topic in the paper (certainly a description of how the numbers were chosen, unless I missed one that's already there). Finding a way to make the measurements appear as strong as they fundamentally are is important in advocating for future measurements.

My more detailed comments, below, mainly related to notation. For most of these my

aim has been to, where possible, avoid confusion for the more casual reader familiar with the notation of the Rodgers-like retrieval community. Obviously your discussion involves many more terms than are typical in the optimal estimation-based approaches, so some clash is perhaps inevitable. However, it would be prudent to avoid them where possible.

Another common thread in my more minor comments is the author's use of the word "accuracy" to describe quantities that I believe are more commonly referred to as "precision". In my experience, "accuracy" is used to described non-random (at least in physical origin if not geophysical-product consequence) terms that affect the measurement, such as errors in instrument calibration and spectroscopy. Precision relates to the uncertainty introduced by "noise" in the radiance observations, and is, I believe, the correct term for the errors the authors are aiming to characterize here.

Finally, the figures, while arguably numbered logically are not introduced in the text in that order. The current order has 3 first then, 6, 1a, 4, 5, then 2. I'm fine with keeping the figures in the order/numbers they currently have, but I would suggest that you arrange for the text to introduce them in that order for consistency. Just having a new sentence up font that introduces them in that order is probably simplest.

Specific comments:

—- Page 1

Line 18: "taken close from one another" -> "taken close to one another"

Line 23: "accuracy" -> "precision", see above.

—- Page 2

Line 3: "quality" is rather a loose term. Consider clarifying it further (e.g., information content, effective resolution, faithfulness to "true" atmosphere, etc.).

Line 4: Suggest you delete "given in equation (2)" as it's a forward reference. Instead

refer to Tikhonov paper?

Line 6: Is the "classic Tikhonov" approach necessarily first order? I know of some teams using second-order Tikhonov routinely.

Line 17: "air radiance" feels like odd wording to me, why not just "radiance"?

—- Page 3

Eq (2) and line 24: Surely if $L_0$ is a term in $S_a^{-1}$ (as opposed to just $S_a$), then what it contains is more like reciprocal standard deviations is it not?

Line 27: I'm not sure that characterizing the alpha terms as "unphysical" is that fair. One could easily scale them with some suitable length term to cast them into constraints on spatial variability (which is more or less what you yourselves are doing). Also, such scaling would arguably render them less "grid dependent".

—- Page 4

Line 19: The term "covariance kernel" was new to me. Is it widely used? Is there some reference for it?

Equation 5: Doesn't $C^{-1}$ need a $_k$ subscript?

—- Page 5

Equation 9: I suggest you split this in two and have $||\phi||^2 = ...$ dU ** <comma> where <newline> ** $\delta(\phi ...$, and call these two separate equations.

Line 23 and 25: Not sure how, if $L_h$ is constant (line 23) you can have "different correlation lengths...". Are you implying $L_h$ only has to be horizontally homogeneous, not vertically constant?

—- Page 6

—- Page 7

Line 13: I don't understand why you're using y in the equation on this line rather than x, as in equation 1. I'd avoid using y for anything state-space related if possible to avoid confusion for the Rodgers-based reader.

Equation 15 and line 30: Why \phi(r_i) in one and \phi(x_i) in the other, why not have them both the same? Of the two, I'd pick r_i, as being less confused with the state vector.

Line 29: bold "a" subscript needed on D

—- Page 8

Line 29: I think that either x_i, x_j's here should either not be bold, as they are components of r_i or r_j, or you mean to actually use r_i and r_j themselves. Same applies for x_l and x_k in line 30.

—- Page 9

Line 26: Suggest "accuracy" -> "precision" as discussed above.

—- Page 10

Line 10: I'd suggest a different letter than y to again avoid confusing the Rodgers community (and those who remember your equation 1!).

Line 28: I like the term "Precision matrix" because, as with the word itself, "more" implies "better". However, is it in common use? Also, I believe it is often referred to as the "Fisher information matrix", but I'm not sure that in all cases $Sa^{-1}$ is indeed the same a the Fisher matrix (I'm stretching beyond my expertise here).

—- Page 11

Discussion starting around line 11, and through the remainder of the section: It's not clear to me how the matrix M in the lines 11-16 relates to the matrix A in equation 22 and beyond. Also, I don't believe A is the Rodgers-style "Averaging kernel" matrix in

this context is it (as Rodgers' A is not s.p.d.)? Again, I'd suggest using a different letter than A. If is intended to be a measure of S_a, as the discussion on Line 30 (page 11) seems to imply, then why not use S_a itself, perhaps with some suitable diacritic (a tilde, or hat or something)? Apologies if I'm missing a critical point here.

—- Page 12

—- Page 13

—- Page 14

Line 16: "my" -> "by" Line 18: Consider "longer" (or "greater") rather than "higher" for vertical correlation length.

—- Page 15

Line 16: Suggest "lower" -> "shorter"

—- Page 16

—- Page 16

—- Page 18

Line 17: Add "operator" after "Jacobian" (at first I'd thought it was meant to be a Jacobian matrix, and wondered why you'd not referred to it as K, as Rodgers does).

—- Figure 3

I suggest that, for the bottom row, as with figure 4, you use a red/blue (or other anti-symmetric) color scale. Also "smoothened" in caption should be "smoothed" I think.
* * *

---

## Referee Comment (RC2) · Anonymous Referee #2 · 27 Sep 2018

The manuscript deals with 3D tomographic inversions of limb sounding data. This is a highly challenging task. Standard approaches must be adopted or replaced in order to handle the extensive calculations required. The manuscript deals mainly with the following important issues

- How to construct the regularisation matrix (denoted as the precision matrix by the authors).

- Usage of irregular grids is explored.

- Monte Carlo estimation of the retrieval precision.

[Figure]

The manuscripts clearly fulfills the basic requirements. The measurements of concern are inside the scope of AMT. The issues considered are relevant and the manuscript includes a significant amount of novel material. However, the presentation and some details should be improved.

General comments

- It is not clear in all parts if the material presented is new or not. It seems that references are lacking. For example, my interpretation of Sec 2.3 is that Delaunay triangulation is claimed to be introduced for this type of measurements. If correct, this should be stated clearly, and also be mentioned in the abstract. On the other hand, in the abstract it seems that this approach is quite standard (which is clearly not the case). Further, Delaunay triangulation has at least been used in other types of radiative transfer simulations, e.g. astronomy. Some references clarifying all this should be included. Further examples are given below.

- It should be clarified to what extent the new approaches are relevant for the more standard 1D and 2D inversions. Most importantly, can 1D and 2D regularisation matrices be constructed in the same manner?

- The new way to construct the regularisation matrix is presented as an extension of Tikhonov regularisation, but I rather see it as a way to approximate the precision matrix of Equation 1? In any case, the approach bridges the gap between Bayesian and Tikhonov regularisation. This is important and should be stressed (depending a bit on if your approach works for 1D and 2D). To be clear, you argue that the simplest way to set the Tikhonov regularisation matrix is to consider statistics of the atmosphere. You use correlation structures, but that approach is essentially identical to the Bayesian approach. That is, you basically argue that the Bayesian approach is to prefer. I am trying to provoke here, on purpose,

to encourage you to extend the discussion and clarify the nice link you provide between Bayesian and Tikhonov regularisation.

- The nomenclature should be revised. Some examples are given below.

General comments on some manuscript parts

**Title:** Should be changed. The present title is too generic, especially considering that the manuscript doesn't involve any real observations. The manuscript deals only with technical improvements of the retrieval step. On the other hand, I don't see any reason to make a restriction to "limb sounding" and "airborne", the new methods are relevant for any 3D observations (also 1D and 2D?).

**Abstract:** I find the abstract vague. See comment about Delaunay triangulation above. Some hard facts would be nice. For example, what reduction in the number of grid points was achieved?

**Section 2:** This section is hard to digest. First of all, it contains relatively advanced mathematics. I must confess that the mathematics in some parts is above my knowledge level, and I must leave it to others to check the details. Further, the presentation is relatively lengthy and is in large parts of textbook character. This makes it hard to distinguish the purely novel contributions and core points, from background information. My suggestion is to make Section 2 more condensed and move the details to an appendix.

As a concrete example, despite Section 2.2 is detailed it is not yet clear to me, after several readings, how the result of Equation 11 shall be used to actually construct the regularisation matrix. Maybe I miss something obvious but I don't see how the result of Eq 11 shall be used to generate $S_a^{-1}$.

The nomenclature should also be revised. For example, the symbol used on the left-hand side of Equation 11 is not defined. Or rather, I assume it's should the same

as in Equation 9. Further, it's unlucky to use $y$ in Equation 14, as $y$ represents the observation in Equation 1.

With respect to Eq.2 and related text: I don't know how Tikhonov formulated the approach (and it was introduced independently by others), and your reference may be correct. On the other hand, I don't think it is fair to say that Tikhonov regularisation is today restricted to consider the first derivative. For example, the description of Tikhonov regularisation in "Numerical recipes" says "... measures of smoothness that derive from first or higher derivatives." An example from the atmospheric field where the second derivative was considered:

Steck, Tilman. "Methods for determining regularization for atmospheric retrieval problems." Applied Optics 41.9 (2002): 1788-1797.

The text can be interpreted as that considering the second derivative is novel, which is not true.

**Section 3:** This section has similar problems as Sec 2. At least the first part dealing Cholesky decomposition contains mainly rather well-known facts (but no reference is given). Do the authors present something new in the part based on graph theory? Anyhow, most of the details can be removed as the final conclusion is that Cholesky decomposition cannot be used. Focus on the final conclusion, found on page 11, lines 7-10.

Also, the next part (page 11, lines 11-30) can be made shorter. First of all, the comments around $Mx = b$ just confused me. As I understand it, all the equations are taken from Allen et al (2000). Then no need to repeat the details, just explain that the solution of Eq 22 has the desired properties. On the other hand, it must be explained how Eq 22 shall be used practically. My understanding is that $A$ shall be set to $S_a$. But we just have its inverse! Can this be clarified by just changing the notation (i.e. replace $A$ with the symbol used elsewhere for what the matrix actually represents).

**Section 4.1:** Title and content do not agree. The section deals also with the forward model.

**Section 4.3:** The vertical correlation length is estimated from horizontal correlation. This seems backward as there should exist much better estimates of the vertical correlation. The later can simply be derived from e.g. sonde data. Take this is a comment, I don't demand a change here.

**Section 4.4:** A demonstration of the new features is, of course, nice to see, and maybe even a demand. However, showing results for a single retrieval case does not prove much. Statistics of an ensemble of retrievals are required to judge if one retrieval is better than another one. For a single case, specifics of the case can make the poorer method to look better. Further, it is also very unclear how "optimal" the regularisation weights used in A actually are? Anyhow, can really optimal weights be found by manually tuning? There are in fact objective methods for setting the weights.

Some of the new features can be tested in a more direct manner, compared to doing full retrievals. For example, a basic demand when selecting a grid is that discretization errors are kept at a sufficiently small level. That is, for me, the first test when introducing a new grid scheme (here Delaunay) is to simply compare forward model simulations and check that results only change in a tolerable way. This test is most critical for D.

By the way, is the same simulated measurement inverted in A to D (presumably based on A)? If a new simulated measurement is done for each case, then a possible discretization errors are swept under the carpet.

In my mind, the most interesting question in the manuscript is how well the calculations actually manage to estimate the exponential covariance assumed (Equation 7)? Is it possible to derive/estimate the $S_a$ implied by the derived $S_a^{-1}$, and check how well the obtained $S_a$ follows the start assumptions? Either for a sub-volume or a smaller test case. My interpretation of Section 2.4 is that you ensure $S_a^{-1}$ to be positive definite and it should then be invertible (for a reasonable large case).

**Conclusions:** Should be extended a bit. Are all problems solved? Or something lacking to attack real observations? Are the new methods applicable in other cases (such as 1D and 2D)?

Details

Page 1, line 8, and elsewhere: It should be considered how the word "accuracy" is used. Accuracy equals systematic error or at least includes this term. See https://en.wikipedia.org/wiki/Accuracy_and_precision. There is no discussion of systematic errors in the text and I think that the word "precision" in general is more proper.

Page 1, line 20: Seems reasonable to reference some 2D retrievals.

Page 1, lines 22-23: This is not a specific 3D issue.

Page 2, line 3: The choice of regularisation constraint does affect the output of the retrieval, but I don't agree that it changes the quality. The regularisation methods are mathematical tools, and, assuming that there is no numerical issues or similar problems, they simply optimize what you have told them to do. That is, if you change regularisation constraint, you select to optimize another metrics, and the result will differ. But can it be claimed generally that one metrics is better than another one? I would say that it depends on the application.

Sec 2.3: I assume you are using some kind of external library to derive the Delaunay triangulation. Which one should be specified? Any other libraries that should be mentioned?

Page 8, line 3: A atmospheric state has been defined as $x$ ($y$ represents the measurement).

Page 9, line 26 and page 11, line 14: "rather difficult" is used in both places. A very vague formulation, be more specific.

Page 9, line 26: I don't agree that this is in general a difficult problem. As you point out, Rodgers (2000) explains how it should be done. Basically, all retrievals come with an error estimation, so it is, in general, a feasible task.

Page 10, line 1: Estimating the retrieval accuracy is not only "valuable", I would say that retrievals without an error estimation should not be used at all. That is, the error estimation is mandatory.

Page 10, line 10: Don't use $y$ to denote a state vector. $y$ denotes the measurement (Eq 1).

Page 11, line 18: "s.p.d." is not defined (I can guess what it means, but that is not good enough).

Page 13, line 15: The figures shall be introduced in order. You start with Figure 3.

Page 16, lines 3 and 13: Join these two comments about weights, to more clearly describe what has been done.

Page 17, line 14: Significant can either be interpreted as "by a large amount" or in a statistical sense. Neither seems to fit in how "significantly" is used here. It suffices to say that there is a 6% increase.

Page 18, lines 30-33: I don't follow the explanation. Why can't you compare apples with apples? If this is not possible, there is little value in the exercise.

---

## Author Response (AR1)

We thank the reviewers for their positive reception of our work, insightful comments, and valuable suggestions that helped to improve this paper.

The replies to both reviews are given below. We do not discuss small technical or typesetting remarks and typos spotted by reviewers here, those were simply applied as recommended. The original referee comments are indented, excerpts from the revised version of the paper are given in italic.

**1 Reply to Referee 1**

**1.1 High level comments**

> My only high-level concern is that the end results, based on simulations for GLORIA, do not necessarily show the instrument or technique off at its best. In part, as I see it, this is because of the tight nature of the a priori constraints imposed, specifically the 0.645 K a priori precision estimate for temperature. As an aside, although the choice of the associated correlation lengths is well described in the manuscript, I was unable to find discussion of how the 0.645K number was arrived at (apologies if I missed it). A consequence of this very tight constraint is that the instrument has to "work very hard" to add useful information, as witness the similarity between the magnitude of the fields in figures 6 and 4 to those in the bottom row of figure 3. In other words, the instrument hasn't been "allowed" to tell you that much more than you already knew.

The following explanation was added to section 4.3: *Our choice of $\sigma$ value for temperature, in particular, may seem rather low. This is related to the choice of a priori: smoothed ECMWF data was used instead of climatologies that would have been more typical in this case. Such an a priori can be expected to match the large scale structures of real temperature more closely, thus reducing the forward model errors in poorly resolved areas and improving the results. It is particularly useful for resolving fine structures, such as gravity waves, which were the main scientific interest of the measurment flight used as a basis for test retrievals [Krisch et al., 2017]. A likely closer match between a priori and retrieval thus requires a lower $\sigma$ value, that, in this setup, represents expected strength of gravity wave disturbances, rather than full thermal variability of the atmospheric region in question.*

It must also be noted that the $\sigma$ value is not an hard limit for deviations from a priori. A gravity wave with an amplitude of 3-3.5 K, which is considered to be among 1% of strongest gravity wave events expected in the measured region, was succesfully retrieved with this setup [Krisch et al., 2017].

**1.2 Details**

> Another common thread in my more minor comments is the author's use of the word "accuracy" to describe quantities that I believe are more commonly referred to as "precision".

This was corrected in several places, as suggested.

> Finally, the figures, while arguably numbered logically are not introduced in the text in that order. The current order has 3 first then, 6, 1a, 4, 5, then 2.

The figures and references were rearranged in the revised paper, now order of figure references in the text matches their numerical order. The old Figure 3 hence became Figure 1, bringing it closer to its first reference, and the old Figures 1 and 2, that were strongly related, were joined into a single figure. The color scale of the old Figure 3 was also changed, as suggested.

> Page 2, Line 3: "quality" is rather a loose term. Consider clarifying it further (e.g., information content, effective resolution, faithfulness to "true" atmosphere, etc.).

Adding detail to this, admittedly vague, statement would repeat the ideas at the end of the paragraph (or those in the previous paragraph). Hence the sentence was simply removed.

> Page 2, Line 6: Is the "classic Tikhonov" approach necessarily first order? I know of some teams using second-order Tikhonov routinely.

This statement was made more specific in the revised paper: *In this paper, we introduce an improved, more physically and statistically motivated approach to regularisation, that requires less tuning. It relies on the calculation of both the first spatial derivative and Laplacian of atmospheric quantities, which provide more complete information about smoothness and feasibility of a particular atmospheric state.* We are not aware of anyone else in the atmospheric community using a combination of first order derivatives and Laplacian for 2D or 3D problems.

> Page 3, Line 27: I'm not sure that characterizing the alpha terms as "unphysical" is that fair. One could easily scale them with some suitable length term to cast them into constraints on spatial variability (which is more or less what you yourselves are doing). Also, such scaling would arguably render them less "grid dependent".

One could, of course, use a physical length instead of each $\alpha$ parameter. That would also allow to, for example, change the grid constant of a regular grid and retain the regularisation strength. Such an approach would, however, result in a large number of tunable parameters (especially if derivatives of higher order than one are to be used). Estimating all of them from model or observation data would be difficult without establishing some relations between them. Our approach provides one way to obtain such relations. Also, a regularisation scheme derived in the form of an integral, as in equation (13) of the revised paper, is relatively easy to adopt to different kinds of grids (e.g. irregular). This gives much more "grid independence" than a mere rescaling that one could do while preserving the form of Tikhonov matrices.

> Page 4, Line 19: The term "covariance kernel" was new to me. Is it widely used? Is there some reference for it?

Covariance kernel, also known as covariance function (https://en.wikipedia.org/wiki/Covariance_function) is just a real function that describes a spatial covariance of the (continuum) random variable. In general, if one defines an operator X by means of an integral of a real (or complex) function over the whole space in question, it is common in applied mathematics to call this function an "X kernel". This is why we chose this term here. It was also used in [Lim and Teo, 2009]. A remark about covariance kernel being the same as the covariance function was added to the revised paper.

> Page 4 Equation 5: Doesn't C^{-1} need a _k subscript?

No. The scalar product (and hence the norm) associated with the covariance operator is induced by volume integral of the inverse of the operator, not the inverse of it's kernel. If one would represent a scalar field $\phi(\mathbf{r})$ by a state vector $\mathbf{x}_\phi$ on some grid, and similarly for $\varphi(\mathbf{r})$ and $H\mathbf{x}_\varphi$, then the discrete equivalent to the integral in equation (5) would be $\mathbf{x}_\phi^T \mathbf{S}_a^{-1} \mathbf{x}_\varphi$. Hopefully the more explicit statement about the equivalence between continuous and discrete formulations (equation (6) in the revised manuscript) will make things clearer for the readers.

> Page 5, Line 23 and 25: Not sure how, if L_h is constant (line 23) you can have "different correlation lengths...". Are you implying L_h only has to be horizontally homogeneous, not vertically constant?

The equation (12) (10 in the original manuscript) holds if, as it was stated in lines 20-21, the correlation lengths are functions of altitude only: $L_h = L_h(z)$, $L_v = L_v(z)$. If, in addition, $L_h$ is constant (i.e. $L_h = \text{const.}$, $L_v = L_v(z)$), then the algebra is further simplified, as was described in lines 23-24.

> Page 10, Line 28: I like the term "Precision matrix" because, as with the word itself, "more" implies "better". However, is it in common use?

Precision, the reciprocal of variance, and precision matrix, the inverse of the covariance matrix, seem to be standard statistical terms (https://en.wikipedia.org/wiki/Precision_(statistics)).

> Discussion starting around line 11, and through the remainder of the section: It's not clear to me how the matrix M in the lines 11-16 relates to the matrix A in equation 22 and beyond. Also, I don't believe A is the Rodgers-style "Averaging kernel" matrix in this context is it (as

Rodgers' A is not s.p.d.)? Again, I'd suggest using a different letter than A. If is intended to be a measure of S_a, as the discussion on Line 30 (page 11) seems to imply, then why not use S_a itself, perhaps with some suitable diacritic (a tilde, or hat or something)? Apologies if I'm missing a critical point here.

The notation and organisation of section 3 was found to be confusing by both reviewers. Therefore, the section was significantly reorganised and notation changed. Please refer to the section 3 in the revised manuscript for details.

**2 Reply to Referee 2**

**2.1 General comments**

It should be clarified to what extent the new approaches are relevant for the more standard 1D and 2D inversions. Most importantly, can 1D and 2D regularisation matrices be constructed in the same manner?

Applicability to 1D and 2D retrievals is discussed in the newly added Appendix B.

The new way to construct the regularisation matrix is presented as an extension of Tikhonov regularisation, but I rather see it as a way to approximate the precision matrix of Equation 1? In any case, the approach bridges the gap between Bayesian and Tikhonov regularisation. This is important and should be stressed (depending a bit on if your approach works for 1D and 2D). To be clear, you argue that the simplest way to set the Tikhonov regularisation matrix is to consider statistics of the atmosphere. You use correlation structures, but that approach is essentially identical to the Bayesian approach. That is, you basically argue that the Bayesian approach is to prefer. I am trying to provoke here, on purpose, to encourage you to extend the discussion and clarify the nice link you provide between Bayesian and Tikhonov regularisation.

A remark about the similarity to Bayesian approach was added, and some statements in section 2.2 clarified.

Title: Should be changed. The present title is too generic, especially considering that the manuscript doesn't involve any real observations. The manuscript deals only with technical improvements of the retrieval step. On the other hand, I don't see any reason to make a restriction to "limb sounding" and "airborne", the new methods are relevant for any 3D observations (also 1D and 2D?).

The title was changed to *3D tomographic limb sounder retrieval techniques: irregular grids and Laplacian regularisation.*

Abstract: I find the abstract vague. See comment about Delaunay triangulation above. Some hard facts would be nice. For example, what reduction in the number of grid points was achieved?

Explicit mention of Delaunay triangulation and the achieved reductions in computational cost were added to the abstract. The new version reads: *Multiple limb sounder measurements of the same atmospheric region taken from different directions can be combined in a 3D tomographic retrieval. Mathematically, this is a computationally expensive inverse modelling problem. It typically requires an introduction of some general knowledge of the atmosphere (regularisation) due to its underdetermined nature. This paper introduces a consistent, physically motivated (no ad-hoc parameters) variant of the Tikhonov regularisation scheme based on spatial derivatives of first order and Laplacian. As shown by a case study with synthetic data, this scheme, combined with irregular grid retrieval methods employing Delaunay triangulation, improves both upon the quality and the computational cost of 3D tomography. It also eliminates grid dependence and the need to tune parameters for each use case. The few physical parameters required can be derived from in situ measurements and model data. Tests show that 82% reduction in the number of grid points and 50% reduction in total computation time, compared to*

*previous methods, could be achieved without compromising results. An efficient Monte Carlo technique was also adopted for accuracy estimation of the new retrievals.*

> As a concrete example, despite Section 2.2 is detailed it is not yet clear to me, after several readings, how the result of Equation 11 shall be used to actually construct the regularisation matrix. Maybe I miss something obvious but I don't see how the result of Eq 11 shall be used to generate S−1a.

Explanations of how to use the equation (11) for obtaining the precision matrix have been added. Refer to equations (6) and (14) in the revised manuscript and the paragraphs above them.

> The nomenclature should also be revised. For example, the symbol used on the left hand side of Equation 11 is not defined. Or rather, I assume it's should the same as in Equation 9. Further, it's unlucky to use y in Equation 14, as y represents the observation in Equation 1.

$\Phi$ instead of $\phi$ in equation 11 was indeed a typo. The notation in most of the section 2.4 was revised not to cause confusion with equation (1).

> With respect to Eq.2 and related text: I don't know how Tikhonov formulated the approach (and it was introduced independently by others), and your reference may be correct. On the other hand, I don't think it is fair to say that Tikhonov regularisation is today restricted to consider the first derivative. For example, the description of Tikhonov regularisation in "Numerical recipes" says "... measures of smoothness that derive from first or higher derivatives." An example from the atmospheric field where the second derivative was considered: Steck, Tilman. "Methods for determining regularization for atmospheric retrieval problems." Applied Optics 41.9 (2002): 1788-1797. The text can be interpreted as that considering the second derivative is novel, which is not true.

This statement was made more specific in the revised paper: *In this paper, we introduce an improved, more physically and statistically motivated approach to regularisation, that requires less tuning. It relies on the calculation of both the first spatial derivative and Laplacian of atmospheric quantities, which provide more complete information about smoothness and feasibility of a particular atmospheric state.* We are not aware of anyone else in the atmospheric community using a combination of first order derivatives and Laplacian for 2D or 3D retrievals.

> Section 3: This section has similar problems as Sec 2.

The notation and organisation of section 3 was found to be confusing by both reviewers. Therefore, the section was significantly reorganised and notation changed. Please refer to the section 3 in the revised manuscript for details.

> Section 4.1: Title and content do not agree. The section deals also with the forward model.

The title was changed to *The GLORIA instrument and data processing*

> Section 4.4: A demonstration of the new features is, of course, nice to see, and maybe even a demand. However, showing results for a single retrieval case does not prove much. Statistics of an ensemble of retrievals are required to judge if one retrieval is better than another one. For a single case, specifics of the case can make the poorer method to look better. Further, it is also very unclear how "optimal" the regularisation weights used in A actually are? Anyhow, can really optimal weights be found by manually tuning? There are in fact objective methods for setting the weights.

One of the aims of this paper is to progress towards a robust and standardised regularisation technique, that could be employed to perform 3D retrievals in various atmospheric conditions without much manual effort. The authors are aware of some of the objective methods for setting the weights for the classic Tikhonov approach, but they, so far, did not prove sufficient to achieve this goal. Therefore, adequately testing both regularisation methods on a large and representative set of atmospheric states and performing appropriate statistical analysis on the results was deemed to be beyond the scope

of this paper. A well-understood test case was used as an example instead. Performing statistical analysis of in situ or model data to determine correlation lengths and standard deviations of various atmospheric quantities, as well as further testing of retrieval methods on larger data sets is an option for further work.

> Some of the new features can be tested in a more direct manner, compared to doing full retrievals. For example, a basic demand when selecting a grid is that discretisation errors are kept at a sufficiently small level. That is, for me, the first test when introducing a new grid scheme (here Delaunay) is to simply compare forward model simulations and check that results only change in a tolerable way. This test is most critical for D. By the way, is the same simulated measurement inverted in A to D (presumably based on A)? If a new simulated measurement is done for each case, then a possible discretisation errors are swept under the carpet.

The authors are very grateful for pointing this out, this important part of test retrieval description was indeed overlooked. The following was added to section 4.2: *The same set of simulated measurements was used for all test retrievals described in this paper. These measurements were obtained by running the forward model on a very dense grid (about twice as dense in each dimension as those of the densest test retrievals). This was done to ensure that the discretisation errors in the simulated measurements would be minimal and would not give any one retrieval an advantage (as it could happen, if they were generated on the same grid as this test retrieval).* Also, a new appendix (Appendix C) with a comparison of forward model output in the case of each retrieval was added to the revised paper.

> In my mind, the most interesting question in the manuscript is how well the calculations actually manage to estimate the exponential covariance assumed (Equation 7)? Is it possible to derive/estimate the Sa implied by the derived S−1 a , and check how well the obtained Sa follows the start assumptions? Either for a sub-volume or a smaller test case. My interpretation of Section 2.4 is that you ensure S−1a to be positive definite and it should then be invertible (for a reasonable large case).

We have come up with two ways to test whether the precision matrix constructed as in equation (14) indeed represents the exponential covariance assumption (8). One way is to construct a covariance matrix directly for a relatively small grid, invert it, and compare the result with the directly obtained precision matrix. Matrix norms are poorly suited for such a comparison, since small test cases are prone to significant edge effects (finite differences perform poorly on grid boundaries, among other issues). Therefore, we simply constructed some test "atmospheric states" – Gaussian wave packets – for the small grid and compared their norms based on the two precision matrices obtained in different ways. This comparison was added to the revised paper as an Appendix A.

An alternative approach would be to construct the precision matrix, invert it, and see if the matrix elements satisfy the exponential relation. These results are harder to interpret: covariance matrix elements do not directly enter any retrieval calculations, it is hard to quantify what errors are acceptable in this case. We will therefore just provide such a comparison here.

We use 20x20x20 regular grid with grid constant equal to 1 in each direction. We set $\sigma = 1$ and consider two cases: $L_h = L_v = L = 2$ and $L = 3$. For each case, a precision matrix is obtained as a discretisation of equation (13) for rectilinear grid. Then the precision matrix is inverted to obtain a covariance matrix. One point near the centre of the grid, the reference point, is selected, and then covariances between every grid point and the reference point are read from the covariance matrix and plotted, as a function of distance between grid point and reference point, in Figure 1 of this document. The plots show the theoretical covariance value as a blue line. Also, the points with distance from reference point lower or equal to 7 were used to perform a least squares fit for covariance function of the form $C_k(\mathbf{r}) = \sigma^2 \exp\left(-\|\mathbf{r}\|/L\right)$ with respect to the parameters $\sigma^2$ and $1/L$. The fits are shown as green lines in Figure 2.1. Best fit was achieved with $\hat{\sigma} = 1.05$, $\hat{L} = 1.89$ in the $L = 2$ case and $\hat{\sigma} = 1.04$, $\hat{L} = 2.66$ in the $L = 3$ case. Hence, in these cases, the precision matrix constructed with our method is in good agreement with theoretical assumptions. One can see from the figures, however, that agreement is worse near the edges of the grid. This is not a surprise, since derivatives, upon which the precision matrix construction is based, cannot be properly evaluated at grid boundaries. The results would also look worse for large $L$ values, those would need a larger grid to properly test.

[Figure]

Figure 1: Covariance to the reference point as a function of distance to the reference point for correlation lengths $L = 2$ and $L = 3$. Blue line – theoretical covariance value, green line – least squares fit for exponential covariance.

Conclusions: Should be extended a bit. Are all problems solved? Or something lacking to attack real observations? Are the new methods applicable in other cases

The following was added to conclusion: *At the time of writing, the methods developed in this paper were already in use to process GLORIA limb sounder data.* Better approximations for regularisation parameters could help to improve the retrievals in the future.

**2.2  Details**

Page 1, line 8, and elsewhere: It should be considered how the word "accuracy" is used. Accuracy equals systematic error or at least includes this term.

This was corrected in several places.

Page 1, lines 22-23: This is not a specific 3D issue.

We were not trying to claim that it is. To make this more explicit, the word "typically" was replaced with: *as it often happens with remote sensing retrievals.*

Page 2, line 3: The choice of regularisation constraint does affect the output of the retrieval, but I don't agree that it changes the quality. The regularisation methods are mathematical tools, and, assuming that there is no numerical issues or similar problems, they simply optimize what you have told them to do. That is, if you change regularisation constraint, you select to optimize another metrics, and the result will differ. But can it be claimed generally that one metrics is better than another one? I would say that it depends on the application.

The (admittedly vague) sentence about retrieval quality was removed.

Sec 2.3: I assume you are using some kind of external library to derive the Delaunay triangulation. Which one should be specified? Any other libraries that should be mentioned?

Delaunay triangulation was generated using CGAL (the Computational Geometry Algorithms Library). This statement was added to section 2.3. CGAL was the only major library that needed to be added to JURASSIC2 code to implement the new methods described in this paper. For more information about JURASSIC2 and other libraries used by this software, refer to [Hoffmann et al., 2008, Ungermann et al., 2011, Ungermann, 2013].

Page 9, line 26: I don't agree that this is in general a difficult problem. As you point out, Rodgers (2000) explains how it should be done. Basically, all retrievals come with an error estimation, so it is, in general, a feasible task.

The beginning of the paragraph was condensed to clarify the main point: theoretical derivations for detailed error estimates do exist, but are usually prohibitively computationally expensive for large problems. In these cases, one needs to resort to other means, such as Monte Carlo. The revised formulation: *Estimating the precision of remote sensing data products generated by means of inverse modelling is essential for the users of the final data and also valuable for evaluation and optimisation of the inverse modelling techniques. Detailed quantitative descriptions of data accuracy can be derived in theory (see validation in section 4.6 and Rodgers (2000)) but they are, in case of large retrievals, too numerically expensive to calculate in practice.*

Page 13, line 15: The figures shall be introduced in order. You start with Figure 3.

The figures and references were rearranged in the revised paper, now order of figure references in the text matches their numerical order. The old Figure 3 hence became Figure 1, bringing it closer to its first reference, and the old Figures 1 and 2, that were strongly related, were joined into a single figure.

Page 16, lines 3 and 13: Join these two comments about weights, to more clearly describe what has been done.

Descriptions of the regularisations for retrievals A and B were extended slightly. For the case of A: *[Regularisation weights] are typically tuned ad-hoc (adjusted by trial-and-error until optimal retrieval results can be achieved), validated against model data and, when using real observations, in situ measurements.* For the case of B: *The grid and interpolation methods were identical to retrieval A, but the regularisation was replaced with the second order scheme from equation (12) and the correlation lengths derived in section 4.3 from in situ observations. No subsequent tuning of these parameters was performed.*

Page 18, lines 30-33: I don't follow the explanation. Why can't you compare apples with apples? If this is not possible, there is little value in the exercise.

A Monte Carlo run that can be compared "apples to apples" with the retrieval results shown has been added to the revised manuscript (Figure 5 Ds). This description of the old and newly added Monte Carlo results should clarify what has been done: *
[revised manuscript text omitted]
}_a$. Then we can set  $x = \tilde{\mathbf{L}}u$, and indeed  $\langle xx^T \rangle = \langle \tilde{\mathbf{L}}uu^T\tilde{\mathbf{L}}^T \rangle = \tilde{\mathbf{L}}\langle uu^T \rangle \tilde{\mathbf{L}}^T = \mathbf{S}_a$, i.e. the vector  $x$ has the required covariance (the angle brackets in the expression above denote the expected value).

A widely used technique for obtaining the matrix $\mathbf{L}$ is the Cholesky decomposition. Given the covariance matrix $\mathbf{S}_a = \{s_{ij}\}$ it explicitly provides a lower triangular matrix root $\tilde{\mathbf{L}} = \{l_{ij}\}$ satisfying $\tilde{\mathbf{L}}\tilde{\mathbf{L}}^T = \mathbf{S}_a$ as shown in (23).

$$l_{ij} = \begin{cases} \sqrt{s_{ij} - \sum_{k=1}^{i-1} l_{ik}^2}, & i = j \\ l_{jj}^{-1}\left(s_{ij} - \sum_{k=1}^{j-1} l_{ik}l_{jk}\right), & i > j \\ 0, & i < j \end{cases} \tag{23}$$

In practice we do not usually assemble the covariance matrix $\mathbf{S}_a$, but rather its inverse: the precision matrix $\mathbf{S}_a^{-1}$, because the latter is sparse. This is not an issue, since $\mathbf{S}_a^{-1}$ is then also symmetric positive definite, so one can compute its root $\mathbf{L}$ such that $\mathbf{L}\mathbf{L}^T = \mathbf{S}_a^{-1}$ in the same way as above, and then obtain  $x$ from the linear system  $\mathbf{S}_a^{-1}x = \mathbf{L}u$. We will refer to the components of precision matrix by $\mathbf{S}_a^{-1} = \{a_{ij}\}$.

Cholesky decomposition does not preserve the sparse structure of $\mathbf{S}_a^{-1}$, i.e. the $\mathbf{L}$ obtained in this way will typically have many more non-zero entries than $\mathbf{S}_a^{-1}$. It follows from (23), however, that if $\mathbf{S}_a^{-1}$ has lower half-bandwidth $w$ (definition in equation (24) below), then so has $\mathbf{L}$

$$w = \min\{k \geq 0 : i - j > k \Rightarrow a_{ij} = 0\} \tag{24}$$

In practice, this means that if $\mathbf{S}_a^{-1}$ is a sparse $N \times N$ matrix with half-bandwidth $w \ll N$, $\mathbf{L}$ will have approximately $N(w+1)$ non zero entries. Cholesky decomposition is hence well suited for computing diagnostics of 1D retrieval: assuming that the value of the retrieved atmospheric parameter at one point only directly correlates with its $w$ nearest neighbouring points in each direction, one gets a precision matrix with half-bandwidth $w$ and can compute $\mathbf{L}$ cheaply with $O(Nw^2)$ operations.

The situation is very different in the higher dimensions. We will use some results of graph theory to show that Cholesky decomposition is not practical in those cases. The $n$-dimensional lattice graph $P_k^n$ consists of an $n$-dimensional rectangular grid of size $k$ in each direction, with edge between any two grid neighbours. Let us label the vertices of $P_k^n$ with integers: $P_k^n = \{v_i : 1 \leq i \leq k^n\}$, and let $q$ be the maximum difference of indexes of neigbouring vertices ($q = \max\{|i - j| : v_i, v_j \text{adjacent}\}$). Then the *bandwidth* $\varphi(P_k^n)$ of $P_k^n$ is defined as the minimum possible value of $q$ among all possible ways to label the vertices. Now say we have some physical quantity defined on each vertex of this grid. Then any reasonable precision matrix $\mathbf{S}_a^{-1}$ for this grid would at least give non-zero correlations between grid neighbours, i.e. $\left(\mathbf{S}_a^{-1}\right)_{ij} \neq 0$ if $v_i$ and $v_j$ are neighbours. Then, by comparing the definitions of $\varphi(P_k^n)$ and lower-half bandwidth $w$ of the precision matrix from the last paragraph, one can see that $2w + 1 \geq \varphi(P_k^n)$. FitzGerald (1974) showed that $\varphi\left(P_k^2\right) = k$ and $\varphi\left(P_k^3\right) = \lfloor 3k^2/4 + k/2 \rfloor$. Therefore, the narrowest possible half-bandwidths of $\mathbf{S}_a^{-1}$ are $w = O(k)$ in 2D and $w = O\left(k^2\right)$ in 3D. It follows that if the grid contains a total of $N$

points, the computational cost of Cholesky decomposition would be $O(N^2)$ in 2D and $O\left(N^{7/3}\right)$ in 3D, which is unsatisfactory for large retrievals.

For these higher dimensions, we need to use sparse matrix iterative techniques to reduce computational cost and memory storage requirements, such as Krylov subspace methods.  In general, it is rather difficult to find a simple iteration scheme that would compute a square root of a matrix and converge reasonably fast. Here we will follow an algorithm proposed by Allen et al. (2000). Consider a system of linear ordinary differential equations (ODEs) with initial condition

$$\begin{cases} \mathrm{d}\boldsymbol{v}/\mathrm{d}t = -\frac{1}{2}\left(\mathbf{S}_\mathrm{a}^{-1}t-(1-t)I\right)^{-1}\left(\mathbf{I}-\mathbf{S}_\mathrm{a}^{-1}\right)\boldsymbol{v}(t) \\ \boldsymbol{v}(0)=\boldsymbol{u} \end{cases} \tag{25}$$

where  $\boldsymbol{v}(t)$ is a column vector of size $N$, $\mathbf{I}$ is the identity matrix and  $\mathbf{S}_\mathrm{a}^{-1}$ is a $N\times N$ symmetric positive definite (s. p. d.  matrix, prescaled so that $\|\mathbf{S}_\mathrm{a}^{-1}\|_\infty < 1$ (i. e. $\mathbf{S}_\mathrm{a}^{-1}-\mathbf{I}$ is non-singular).  $(\mathbf{S}_\mathrm{a}^{-1}-\mathbf{I})$ and $(\
[revised manuscript text omitted]

---

## Author Response (AR2)

We thank the reviewer for the comments and suggestions.

The reply is given below. We do not discuss small technical or typesetting remarks and typos spotted by the reviewer here, those were simply applied as recommended. The original referee comments are indented, excerpts from the revised version of the paper are given in italic.

**Reply to Referee 2**

Section 2.2 is still not sufficiently clear. My assumption is that the aim is to calculate the precision matrix, and I then expected to see an equation clearly showing how the elements of the inverse if Sa are calculated. I can not identify such an equation. When reading the new manuscript I get unsure. Is it in fact only the discrete version of Eq. 13 that is evaluated? If yes, is it then not necessary to calculate the derivative of this quantity with respect to x, in order to minimise Eq. 2? I then miss an explanation of how this is done.

The explicit expression for  $\mathbf{S}_a^{-1}$  was added (equations (15) and (16)). Some references throughout the paper were changed to refer, where appropriate, to the new equation (15) instead of its continuous equivalent (13). This explicit expression was previously omitted, because it was thought to be extremely cumbersome and not very illuminating. The revised manuscript clearly states that  $\mathbf{S}_a^{-1}$  is explicitly calculated in our implementation.

The first part of Sec. 3 still totally lacks references. This makes it impossible to judge if there is anything new here. At least parts are of rather a basic character and are discussed in a number of books.

References concerning computational cost scalings for matrix inversion, the use of Cholesky decomposition for Monte Carlo simulations, and the explicit form of this decomposition were added. Those parts of section 3 are indeed of basic character, but are needed further to explain how a matrix root of  $\mathbf{S}_a$  is obtained from  $\mathbf{S}_a^{-1}$ , and to demonstrate why the classic Cholesky approach is not suitable for our 3D retrievals.

I still think that it is not fair to describe older applications of Tikhonov regularisation as restricted to first order derivatives (page 2, line 10). That this is maybe true for 3D is not sufficient argument (as argued in the author response file).

The mention of "first order" was removed.

It is still not clear in the manuscript how the regularisation weights for retrieval A were set. The revised manuscript gives the following explanation. The retrieval weights for retrieval A were tuned ad-hoc, i.e. adjusted by trial-and-error until optimal retrieval results can be achieved, starting from the default values obtained as in [Hansen and O'Leary, 1993]. While tuning, retrieval results are validated against the atmospheric state used to generate the synthetic measurements. When using real observations, results are validated against in situ measurements and model data.

This was slightly reformulated compared to the previous version of the manuscript and the reference added. Tuning is necessary, since the retrieval results with the default (obtained mathematically as in [Hansen and O'Leary, 1993], rather than using physical information) weights typically give clearly suboptimal retrieval results. The authors agree that such an approach is far from ideal, which is the main motivation for work on regularisation described in this paper.

Page 11, line 16: Is "adjacent" a typo? If not, I don't understand. This was not a typo, but perhaps too short, and hence unclear, formulation. "adjacent" was changed to "adjacent vertices".

**References**

[Hansen and O'Leary, 1993] Hansen, P. and O'Leary, D. (1993). The use of the l-curve in the regularization of discrete ill-posed problems. SIAM Journal on Scientific Computing, 14(6):1487–1503.

**3D** tomographic limb sounder retrieval techniques: irregular grids and Laplacian regularisation**

Lukas Krasauskas1, Jörn Ungermann1, Stefan Ensmann1, Isabell Krisch1, Erik Kretschmer2, Peter Preusse1, and Martin Riese1

[revised manuscript text omitted]
}_{\rm apr}\right)^T \mathbf{S}_{\rm a}^{-1} \left(\hat{\boldsymbol{x}} - \boldsymbol{x}_{\rm apr}\right)\right)$$
(3)

This immediately justifies the second term of equation (1), provided that  $S_a^{-1}$  represents, to some extent, the actual statistics of the atmosphere. This is often referred to as the Bayesian approach to regularisation (Rodgers, 2000). The precision matrix

- from equation (2), is, however, a mathematical device not meant to represent the physical world, as discussed in the previous section. Furthermore, we would like to derive a coordinate independent expression for the regularisation term of cost function (also unlike equation (2)), as this would be useful for work with irregular grids and allow us to use the same regularisation on completely different grid geometries. To achieve these goals, we use continuous covariance operators and their associated
- 15 norms and only discretise them as a last step, hence preserving the grid independence and their statistical interpretation. The resulting discrete regularisation is a variant of Tikhonov in its final numerical form, but can be interpreted as a realisation of general continuous covariance relations. The following paragraph introduces some of the required formalism (Lim and Teo, 2009; Tarantola, 2013). Let us denote the departure of the atmospheric quantity *f*(*r*) from the a priori by *φ*(*r*) = *f*(*r*) − *f*apr(*r*) for some *r* ∈ ℝ3. In some finite volume *V* ⊂ ℝ3, we define the covariance operator C by

20
$$C\phi(\mathbf{r}) = \int_{\mathbf{r}' \in V} \phi(\mathbf{r}') C_k(\mathbf{r}, \mathbf{r}') dV$$
 (4)

where  $C_k : \mathbb{R}^3 \times \mathbb{R}^3 \to \mathbb{R}$  is the covariance kernel (also known as covariance function). Then we can treat the scalar fields  $\phi(\mathbf{r})$  as elements of Hilbert space with the product

$$\langle \phi, \varphi \rangle = \int_{V} \phi(\boldsymbol{x}) \operatorname{C}^{-1} \varphi(\boldsymbol{x}) \,\mathrm{d}V$$
(5)

which induces the norm  $\|\phi\|^2 = \langle \phi, \phi \rangle$ . Once the explicit expression for the norm is found it can be discretised, representing 25 the scalar field  $\phi(\mathbf{r})$  by the atmospheric state vector  $\mathbf{x} - \mathbf{x}_{apr}$ . Then we can write

$$\|\phi\|^2 = (\boldsymbol{x} - \boldsymbol{x}_{apr})^T \mathbf{S}_a^{-1} (\boldsymbol{x} - \boldsymbol{x}_{apr})$$
(6)

It now remains to find an appropriate covariance kernel  $C_k$ . Let us first consider an atmosphere that is isotropic and has the same physical and statistical properties everywhere. Then one would expect  $C_k(\mathbf{r}, \mathbf{r'}) = g(||\mathbf{r} - \mathbf{r'}||)$  with some monotonously decreasing function  $g: [0, \infty) \rightarrow [0, \infty)$ . The most common kernel used in literature for fluid dynamics, meteorology and

similar applications is the parametric Matérn covariance kernel (introduced, e.g. by Lim and Teo (2009))

$$C_{\nu}(\boldsymbol{r},\boldsymbol{r}') = \sigma^2 \frac{2^{1-\nu}}{\Gamma(\nu)} \left(\sqrt{2\nu} \frac{d}{L}\right)^{\nu} K_{\nu}\left(\sqrt{2\nu} \frac{d}{L}\right)$$
(7)

where  $d = \|\mathbf{r} - \mathbf{r}'\|$ ,  $\sigma$  and L are the standard deviation and typical length scale of structures (correlation length), respectively, of the atmospheric quantity in question.  $K_{\nu}$  is the modified Bessel function of the second kind,  $\Gamma(\nu)$  is the Gamma function.

5 Exponential and Gaussian covariance kernels are special cases of the kernel (7), with  $\nu = 0.5$  and  $\nu = 1$  respectively. We choose the exponential covariance

$$C_k(\boldsymbol{r}, \boldsymbol{r}') = \sigma^2 \exp\left(-\frac{\|\boldsymbol{r} - \boldsymbol{r}'\|}{L}\right)$$
(8)

as it closely resembles the Matérn with  $\nu$  values that are typically used for fluid problems and allows for analytic derivation of the subsequently required quantities. Also, having a significant number of parameters to estimate theoretically, we could not make a good use of the flexibility provided by the free parameter of the Matérn covariance in any case.

It can be shown (Tarantola, 2013) that the norm associated with the covariance (8) is

10

$$\|\phi\|^{2} = \frac{1}{8\pi\sigma^{2}} \int_{r\in V} \left[ \frac{\phi^{2}}{L^{3}} + \frac{2\|\nabla\phi\|^{2}}{L} + L\left(\Delta\phi\right)^{2} \right] \mathrm{d}V$$
(9)

Neglecting neglecting some boundary terms. In a more realistic picture of the atmosphere, the correlation length L depends on altitude and strongly depends on direction: correlation between vertically separated air parcels is much weaker than between

15 air parcels separated by the same distance horizontally. We propose to deal with anisotropic or variable L by performing a coordinate transformation such that L would be isotropic and constant in resulting coordinates. In particular, let  $U, V \subset \mathbb{R}^3$ and consider a bijective map  $\boldsymbol{\xi} : V \to U$  such that both  $\boldsymbol{\xi}$  and  $\boldsymbol{\xi}^{-1}$  are twice differentiable on their respective domains and have non-zero first and second derivatives everywhere. Then, using integration by substitution and basic vector calculus identities, equation (9) can be written as

$$\quad \|\phi\|^2 = \int_{\boldsymbol{u}\in\boldsymbol{\xi}^{-1}(V)} \frac{|\det(\mathbf{D}\boldsymbol{\xi})|}{8\pi\sigma^2} \left( \frac{(\phi(\boldsymbol{\xi}))^2}{L^3} + \frac{2\|\boldsymbol{\delta}(\phi(\boldsymbol{\xi}))\|^2}{L} + L\operatorname{Tr}^2 \left[ (\mathbf{D}\boldsymbol{\xi})^{-T} \left[ \mathbf{D}^2\phi(\boldsymbol{\xi}) - \boldsymbol{\delta}(\phi(\boldsymbol{\xi})) \cdot \mathbf{D}^2\boldsymbol{\xi} \right] (\mathbf{D}\boldsymbol{\xi})^{-1} \right] \right) \mathrm{d}U \quad (10)$$

$$\boldsymbol{\delta}(\boldsymbol{\phi}(\boldsymbol{\xi})) = (\mathbf{D}\boldsymbol{\xi})^{-T} \nabla(\boldsymbol{\phi}(\boldsymbol{\xi}))$$
(11)

Here  $(D\xi)_{ij} = \partial \xi_i / \partial u_j$  is the Jacobian, and we define the matrix  $(D^2 f)_{ij} = (\partial^2 f) / (\partial u_i \partial u_j)$ .  $\xi$  can be chosen so that all its spatial derivatives could be computed analytically, and  $\nabla(\phi(\xi))$ ,  $D^2(\phi(\xi))$  are the derivatives of  $\phi$  in the transformed space, so numerical evaluation of (10) is similar in complexity to that of (9), the only notable difference being the need to compute the

so numerical evaluation of (10) is similar in complexity to that of (9), the only notable difference being the need to compute the mixed derivatives  $\partial^2 \phi / (\partial u_i \partial u_j)$ ,  $i \neq j$  if D $\boldsymbol{\xi}$  is not diagonal. A large and relevant class of suitable maps  $\boldsymbol{\xi}$  can be defined, for example, by assuming that the correlation lengths in vertical and horizontal direction are smooth functions of altitude  $L_v(z)$ ,  $L_h(z)$  (we use Cartesian coordinates (x, y, z) where z axis is vertical). Then the map

$$\boldsymbol{\xi}^{-1}(x, y, z) = \frac{1}{L} \left( x L_h(z), y L_h(z), \int_0^z L_v(z') dz' \right)$$
(12)

[revised manuscript text omitted]

$$f(\mathbf{r}) = f(\mathbf{r_0}) + \mathbf{k} \cdot (\mathbf{r} - \mathbf{r_0}), \quad \mathbf{k} = \text{const.}$$
(17)

If  $r_i$ ,  $0 \le i \le 4$  are the four vertices of the Delaunay cell, the (unique) constant gradient k can obtained from a system of 3 linear equations

25
$$f(\mathbf{r}_i) = f(\mathbf{r}_0) + \mathbf{k} \cdot (\mathbf{r}_i - \mathbf{r}_0), \quad 1 \le i \le 3$$
 (18)

This method ensures that the interpolated quantity is continuous and consistent with the volume integration scheme described in section 2.5. Implementation is simple and fast, as any point inside a given Delaunay cell can be interpolated with data about that cell only. The gradient  $\nabla f$  of the atmospheric quantity, is, however, generally discontinuous at cell boundaries making the interpolation unsuitable for direct use in spatial derivative evaluation. The Computational Geometry Algorithms Library

30 (CGAL, https://www.egal.org) was used to construct Delaunay triangulations for our grids.

**2.4 Derivatives**

25

In order to evaluate the cost function based on 3D exponential covariance (13, 15), we need to estimate  $\nabla \phi$  and  $\Delta \phi$  at every grid point (as before,  $\phi = f - f_{apr}$  is the departure of atmospheric quantity f from the a priori)-, i.e. construct the matrices  $\mathbf{L}_{i}$ ,  $\mathbf{L}_{ii}$ , i = x, y, z of equation (15). This section describes the algorithm we use to achieve that on irregular Delaunay grids.

5 We begin by establishing some requirements that our derivative-estimation algorithm will have to meet. Firstly, recall that our inverse model assembles the precision matrix S-1a so that (x - xapr)T S-1a (x - xapr) would be a numerical representation of (13) (or, more generally, (10)). Let us write S-1a = A+B, where A is a diagonal matrix representing the first integral of (13) (or first term of (10)), and B represents the remaining terms. If B were an exact representation, it would be positive definite by construction, but this may not hold with ∇φ and Δφ obtained numerically for a finite grid. If indeed an atmospheric state
10 vector q exists so that qTBq = 0, then for every atmospheric state x

$$(\boldsymbol{x} \pm \boldsymbol{q} - \boldsymbol{x}_{apr})^T \mathbf{B} (\boldsymbol{x} \pm \boldsymbol{q} - \boldsymbol{x}_{apr}) = (\boldsymbol{x} - \boldsymbol{x}_{apr})^T \mathbf{B} (\boldsymbol{x} - \boldsymbol{x}_{apr}) \pm 2 (\boldsymbol{x} - \boldsymbol{x}_{apr})^T \mathbf{B} \boldsymbol{q}$$
(19)

hence one of the states  $x \pm q$  would be favoured (have a smaller cost function) over x by the inverse modelling algorithm. Therefore, a scaled version of q would appear on any retrieval as noise. This particular type of noise would only be suppressed by the cost function term from  $(x - x_{apr})^T \mathbf{A} (x - x_{apr})$ . This guarantees only that x is not very far from a priori, but does nothing to ensure that it is even continuous. This often results in retrievals with unphysical periodic structures (noise) and is

15 nothing to ensure that it is even continuous. This often results in retrievals with unphysical periodic structures (noise unacceptable.

We deal with this problem by explicitly ensuring that **B** is positive definite. Consider the estimation of derivatives at a single grid point  $a \in \mathbb{R}^3$ .  $\nabla \phi(a)$  and  $\Delta \phi(a)$  are numerically estimated from the values of  $\phi$  in some (say m) grid points near a. We can write this as a map

$$\quad D_{\boldsymbol{a}}(\phi) : \mathbb{R}^m \to \mathbb{R}^6, \quad \{\phi(\boldsymbol{r}_i) - \phi(\boldsymbol{a}) : 1 \le i \le m\} \mapsto \left\{\frac{\partial \phi}{\partial x}, \frac{\partial \phi}{\partial y}, \frac{\partial \phi}{\partial z}, \frac{\partial^2 \phi}{\partial x^2}, \frac{\partial^2 \phi}{\partial y^2}, \frac{\partial^2 \phi}{\partial z^2}\right\}$$
(20)

Now  $q^T \mathbf{B} q = 0$ ,  $q \neq \mathbf{0}$  implies that at every grid point  $\mathbf{
[revised manuscript text omitted]
(\mathbf{r}) = \frac{1}{4} \sum_{i=0}^{3} f(\mathbf{r}_i) + \mathbf{k} \cdot \left(\mathbf{r} - \frac{1}{4} \sum_{i=0}^{3} \mathbf{r}_i\right)$$
(23)

Now choose Cartesian coordinates (x', y', z') where  $\mathbf{r_0} = (H, 0, 0)$  and  $\mathbf{r}_i, 1 \le i \le 3$  lie in the plane x' = 0 and consider the x' component of the integral  $\mathbf{I} = \int_V \left(\mathbf{r} - 1/4\sum_{i=0}^3 \mathbf{r}_i\right) \mathrm{d}V$

10
$$I_{x'} = \int_{V} \left( x' - \frac{H}{4} \right) dV = \int_{0}^{H} x' S_0 \left( \frac{H - x'}{H} \right)^2 dx' - \frac{HV}{4} = \frac{H^2 S_0}{12} - \frac{HV}{4} = 0$$
 (24)

As we can choose such coordinates for any vertex instead of  $r_0$ , the integral I has zero component in 3 linearly independent directions, so I = 0. Then integrating (23) over V yields (22) and completes the proof.

[revised manuscript text omitted]

For these higher dimensions, we need to use sparse matrix iterative techniques to reduce computational cost and memory 5 storage requirements, such as Krylov subspace methods. In general, it is rather difficult to find a simple iteration scheme that would compute a square root of a matrix and converge reasonably fast. Here we will follow an algorithm proposed by Allen et al. (2000). Consider a system of linear ordinary differential equations (ODEs) with initial condition

$$\begin{cases} \mathrm{d}\boldsymbol{v}/\mathrm{d}t = -\frac{1}{2} \left( \mathbf{S}_{\mathrm{a}}^{-1} t - (1-t) I \right)^{-1} \left( \mathbf{I} - \mathbf{S}_{\mathrm{a}}^{-1} \right) \boldsymbol{v}(t) \\ \boldsymbol{v}(0) = \boldsymbol{u} \end{cases}$$
(27)

where  $\boldsymbol{v}(t)$  is a column vector of size N,  $\mathbf{I}$  is the identity matrix and  $\mathbf{S}_{a}^{-1}$  is a  $N \times N$  symmetric positive definite (s. p. 10 d.) matrix, prescaled so that  $\|\mathbf{S}_{a}^{-1}\|_{\infty}
- 20 from several points along the flight path and use tomography to improve resolution in the horizontal direction perpendicular to the flight path. Best results with tomographic retrievals are achieved, however, when the aircraft flies in a path close to circular (hexagonal flight paths are used in practice, so that the aircraft could fly straight most of the time) of around 400 km in diameter, and GLORIA observes the air masses in the middle from many directions (Ungermann et al., 2011). A horizontal resolution down to 25 km in both directions can be achieved this way. As any limb observer, GLORIA can only provide tangent point
- 25 data about air masses at and below aircraft flight altitude, which is limited to, approximately 20 km for the M55 Geophysica and 15 km for HALO. For more detail on GLORIA, refer to e.g. Friedl-Vallon et al. (2014).

The implementations of the algorithms described in the sections 2 and 3 were integrated into the Jülich Rapid Spectral Simulation Code Version 2 (JURASSIC2). The forward model of atmospheric radiance used in this code employs a forward model for atmospheric radiation based on radiances obtained from the emissivity growth approximation method (Weinreb

30 and Neuendorffer (1973), Gordley and Russell (1981)) and the Curtis–Godson approximation (Curtis (1952), Godson (1953)). For more information about JURASSIC2, refer to Hoffmann et al. (2008); Ungermann et al. (2011); Ungermann (2013). The performance of the new algorithms, both in terms of output quality and computational cost, is evaluated in the rest of this section.